# Ligand-directed two-step labeling to quantify neuronal glutamate receptor trafficking

Kento Ojima[1], Kazuki Shiraiwa[1], Kyohei Soga[2], Tomohiro Doura[2], Mikiko Takato[1], Kazuhiro Komatsu[1], Michisuke Yuzaki [3], Itaru Hamachi [1✉] & Shigeki Kiyonaka [2✉]

The regulation of glutamate receptor localization is critical for development and synaptic plasticity in the central nervous system. Conventional biochemical and molecular biological approaches have been widely used to analyze glutamate receptor trafficking, especially for α-amino-3-hydroxy-5-methyl-4-isoxazole-propionate-type glutamate receptors (AMPARs). However, conflicting findings have been reported because of a lack of useful tools for analyzing endogenous AMPARs. Here, we develop a method for the rapid and selective labeling of AMPARs with chemical probes, by combining affinity-based protein labeling and bioorthogonal click chemistry under physiological temperature in culture medium. This method allows us to quantify AMPAR distribution and trafficking, which reveals some unique features of AMPARs, such as a long lifetime and a rapid recycling in neurons. This method is also successfully expanded to selectively label N-methyl-D-aspartate-type glutamate receptors. Thus, bioorthogonal two-step labeling may be a versatile tool for investigating the physiological and pathophysiological roles of glutamate receptors in neurons.

[1] Department of Synthetic Chemistry and Biological Chemistry, Graduate School of Engineering, Kyoto University, Kyoto 615-8510, Japan. [2] Department of Biomolecular Engineering, Graduate School of Engineering, Nagoya University, Nagoya 464-8603, Japan. [3] Department of Physiology, School of Medicine, Keio University, Tokyo 160-8582, Japan. ✉email: ihamachi@sbchem.kyoto-u.ac.jp; kiyonaka@chembio.nagoya-u.ac.jp

In the central nervous system, ionotropic glutamate receptors (iGluRs) mediate fast excitatory neurotransmission. iGluRs are categorized into distinct classes based on their pharmacology and structural homology, including the α-amino-3-hydroxy-5-methyl-4-isoxazole-propionate (AMPA) receptor (GluA1–4), kainate receptor (GluK1–5), N-methyl-D-aspartate (NMDA) receptor (GluN1, GluN2A–D, GluN3A–B), and δ receptors (GluD1–2)[1]. iGluRs assemble as tetramers, and functional receptors are formed exclusively by the assembly of subunits within the same functional receptor class.

AMPA receptors (AMPARs), which are mainly permeable to monovalent cations ($Na^+$ and $K^+$), mediate the majority of excitatory synaptic transmission. AMPARs can form homo-tetramers or heterotetramers, and subunit compositions are dependent on brain regions. In hippocampal CA1 neurons, the majority of AMPARs are made up of GluA1/A2 and GluA2/A3 subunit combinations, with a small contribution of GluA1 homomers[2,3]. Recent studies have revealed that AMPARs are constitutively cycled in and out of the postsynaptic membrane through endocytosis and exocytosis. The precise regulation of this process is critical for synaptic plasticity, which is the basis of learning, memory, and development in neural circuits[2,3]. Although AMPARs and kainate receptors are activated by glutamate binding, NMDA receptors (NMDARs), which have high permeability to $Ca^{2+}$, require depolarization as well as agonist binding for their activation. Functional NMDARs require the assembly of two GluN1 subunits together with either two GluN2 subunits, or a combination of GluN2 and GluN3 subunits[1]. An NMDAR-dependent $Ca^{2+}$ influx triggers intracellular signal transduction cascades, and the precise targeting of NMDARs to synapses is essential for controlling neuronal connectivity or neuroplasticity[4]. Thus, to understand the molecular mechanisms of learning and memory, it is critical to analyze the membrane localization and trafficking of iGluRs.

Biochemical approaches, such as surface biotinylation assays or related methods, have been widely used to analyze membrane protein localizations, and these methods have been successfully applied to AMPARs[57]. Although they are powerful tools for the analysis of AMPAR trafficking, cell-surface proteins are randomly labeled with biotin using these methods. As a result, purification of biotin-labeled AMPAR is required, which hampers quantitative analyses of trafficking. In contrast, to selectively visualize glutamate receptors, fluorescent proteins are fused to the receptors using genetically encoded approaches. For example, a pH-sensitive variant of GFP (super-ecliptic pHluorin [SEP]) can be fused to the extracellular region of receptors to visualize cell-surface receptors in live neurons[8,9]. Instead of fluorescent proteins, protein tags such as SNAP- or Halo-tags are fused to the receptors for the covalent labeling of small chemical probes at the time that the probes are added[10–12]. The downsizing of these protein tags has been successfully demonstrated by using a short peptide tag (1–3 kDa) and its probe pair[13–15]. More recently, genetic code expansion in combination with bioorthogonal click chemistry has been reported for the fluorescent labeling of iGluRs in HEK293T cells, in which chemical probes are covalently attached to the side chain of an unnatural amino acid residue[16,17]. These genetically encoded approaches have been widely used in trafficking studies of iGluRs, especially for AMPARs. However, in most cases, these methods largely rely on the overexpression of target iGluR subunits. Given the formation of heterotetramers consisting of different subunits in iGluRs, the overexpression of a single iGluR subunit may interfere with the localization and/or trafficking of native iGluRs in neurons. Ideally, endogenously expressed iGluRs should be tagged with small chemical probes[18,19].

In situ chemical protein labeling is ideal for analyzing native proteins in live cells. Affinity-based protein labeling is a powerful technique for the selective modification of target proteins[20–26]. As a traceless affinity-based labeling method for cell-surface proteins, our group has reported ligand-directed acyl imidazole (LDAI) chemistry[24,25]. With this technique, small chemical probes including fluorophores are covalently attached to nucleophilic amino acid residues located near the ligand-binding site. Recently, we have developed an AMPAR-selective LDAI reagent, termed "chemical AMPAR modification 2" (CAM2) reagents, which allows us to label chemical probes to AMPARs endogenously expressed in cultured neurons or acutely prepared brain slices[26]. Although this technique is powerful for the selective modification of chemical probes to AMPARs, there are some restrictions for visualizing or analyzing cell-surface AMPARs. First, live cells need to be kept at low temperatures (e.g., 17 °C) during CAM2 labeling (1–4 h) to suppress the internalization of labeled AMPARs[27]. Second, the neuronal culture medium needs to be exchanged for serum-free medium or buffered saline during labeling to decrease non-specific labeling of serum proteins such as albumin. The relatively long-term exposure (1–4 h) to these non-physiological conditions may interfere with neuronal activity or survival[28–30]. Ideally, neurons should be kept under physiological conditions during chemical labeling.

Here, we show a method for the rapid and selective labeling of AMPARs under physiological temperature in culture medium by combining LDAI-based protein labeling and the inverse electron demand Diels–Alder (IEDDA) reaction, a form of fast click chemistry[31–33]. This two-step labeling allows the quantitative analyses of distribution and/or trafficking of endogenous AMPARs from short to long periods in cultured neurons. In addition, we successfully apply this technique to chemically label and study the trafficking of endogenous NMDARs in neurons.

## Results

**Rapid labeling of surface AMPARs by bioorthogonal two-step labeling**. We propose a bioorthogonal two-step labeling technique, which combines LDAI-based protein labeling with the bioorthogonal IEDDA reaction for the rapid and selective modification of chemical probes to cell-surface iGluRs. For the first step, a strained alkene is covalently attached to iGluRs using LDAI chemistry, where the acyl substitution reaction to nucleophilic amino acid residues is facilitated by selective ligand–protein recognition (first step in Fig. 1a). Next, the labeled alkene group is rapidly modified with tetrazine-conjugated probes (Tz-probes) on the cell surface, as a result of the high selectivity and high reaction rate of the IEDDA reaction (second step in Fig. 1a).

For the selective labeling of a strained alkene to AMPARs in the first step, we designed a CAM2 reagent bearing trans-cyclooctene (TCO), which we termed CAM2(TCO) (Fig. 1b). TCO was selected as the strained alkene because of its extremely fast cycloaddition kinetics in the IEDDA reaction. Compared with the original CAM2 reagents (e.g., CAM2(Ax488)) that bear aromatic fluorophores (see Supplementary Fig. 1a, b), an ethylene glycol linker is added between the reactive acyl imidazole unit and the TCO group in CAM2(TCO) to increase its hydrophilicity. Hydrophobic or aromatic groups have high affinity to albumin abundantly contained in serum[34]; therefore, this improvement decreases the undesired labeling of albumin, which allows the chemical labeling of AMPARs to be conducted in cell culture medium containing serum or substitutes. In addition, the first labeling is conducted at a physiological temperature (37 °C). Although some of the labeled AMPARs are likely to be internalized in this condition, this is not problematic with the two-step labeling technique. This is because the chemical probes are selectively tethered to cell-surface AMPARs in the second step reaction (Fig. 1d).

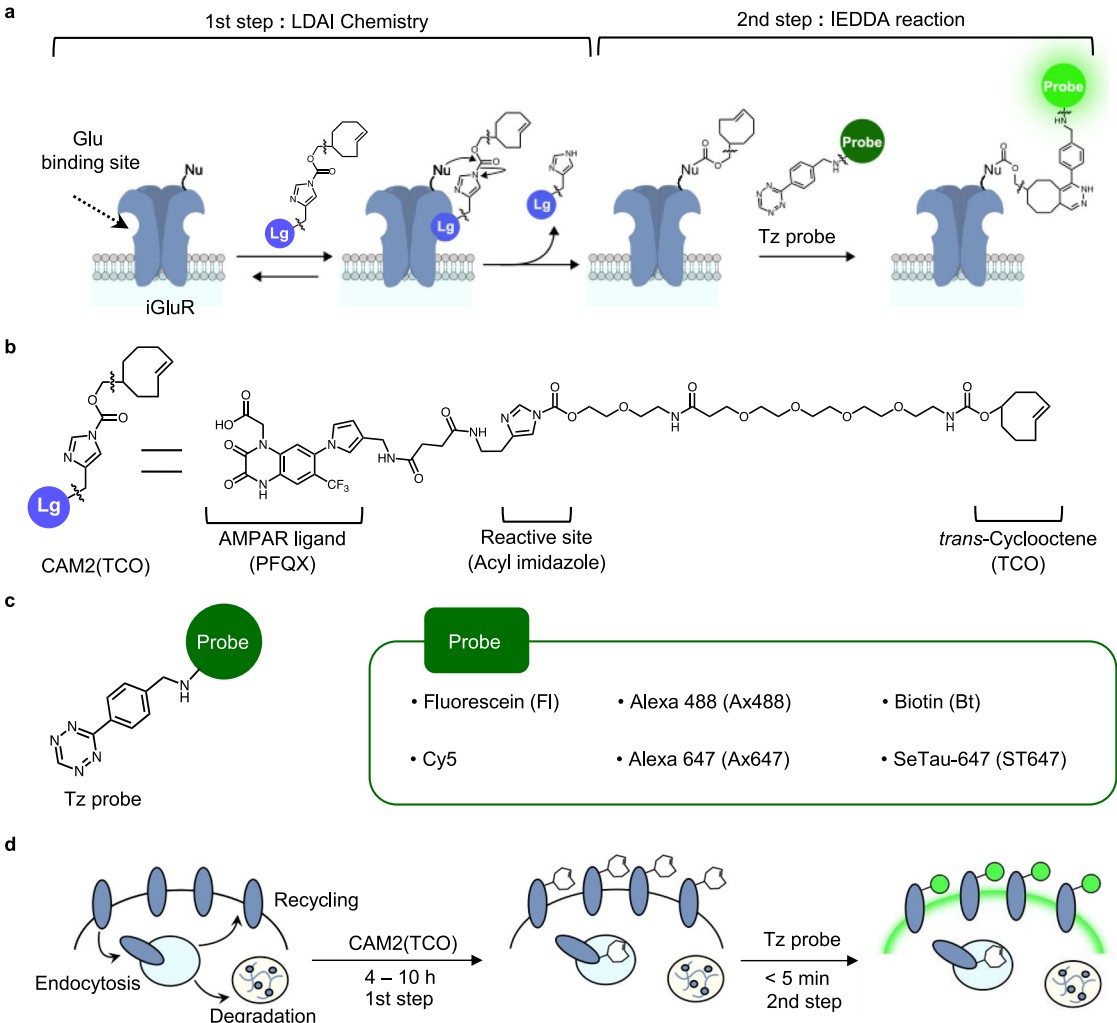

**Fig. 1 Rapid labeling of cell-surface iGluRs by ligand-directed two-step labeling. a** Schematic illustration of the two-step labeling to iGluRs. In the 1st step, a strained alkene is covalently attached to iGluRs by LDAI chemistry. In the 2nd step, Tz probe is selectively tethered by IEDDA reaction. Lg, selective ligand for iGluRs; Nu, nucleophilic amino acid residue. Glu, glutamate. **b** Chemical structure of CAM2(TCO). **c** Chemical structure of Tz probes. The detailed chemical structures are shown in Supplementary Fig. 2. **d** Schematic illustration of the two-step labeling in live cell.

Regarding the second step (the IEDDA reaction), the reaction rate is highly dependent on the chemical structure of the tetrazine group. We selected monoaryl tetrazine, which has both a fast reaction rate and high bioorthogonality, and prepared cell-impermeable Tz-probes bearing hydrophilic and anionic fluorophores or biotin for cell-surface labeling (Fig. 1c and Supplementary Fig. 2). Shortening the reaction time of the probe labeling not only contributes to cell-surface specific labeling, but also decreases the adsorption of the chemical probes to cells, culture dishes, or coverslips. Moreover, some tetrazine-fluorophore conjugates have a "turn-on" response upon the IEDDA cycloaddition[35–37], which contributes to a high signal-to-noise (S/N) ratio in fluorescence imaging.

**Chemical labeling of surface AMPARs ectopically expressed in HEK293T cells**. The designed two-step labeling method was initially examined in HEK293T cells transiently expressing GluA2, a main subunit of AMPARs. For the first step reaction, CAM2(TCO) was added to the culture medium, which included 10% fetal bovine serum (FBS), and the culture dish was incubated at 37 °C for 4 h. The second step reaction was performed for 5 min by adding membrane-impermeable Tz(Fl) for fluorescein

labeling on the cell surface. As shown in Fig. 2a, western blotting of the cell lysate using anti-fluorescein (anti-Fl) antibodies showed a strong band around 110 kDa (lane 1). This band was not observed in the cells co-treated with a competitive ligand (NBQX) or in any other control conditions (lanes 2–5). With regard to the molecular weight of the labeled band, the anti-Fl signal corresponded to the highest signal among multiple bands that were detected using anti-GluA2/3 antibodies (Fig. 2b). The multiple GluA2 bands converged into a single lower band after treatment with peptide-$N$-glycosidase F (PNGase F), which is consistent with previous reports showing that GluA2 is highly glycosylated with $N$-linked sugars[38]. Importantly, in the PNGase F-treated samples, the shifted anti-GluA2 band merged with the anti-Fl signal (Fig. 2b). These findings indicate that highly glycosylated GluA2 is selectively labeled using our methods. Furthermore, in the case of direct fluorescein labeling using the original CAM2(Fl) under the same conditions (see Supplementary Fig. 1b for its structure), there was a strong band around 70 kDa as well as the 110 kDa band (lanes 6–7 in Fig. 2a and Supplementary Fig. 3). The 70 kDa band, whose intensity did not change even in the presence of NBQX, corresponds to albumin contained in serum (for details, see Fig. 2a legend). These results therefore indicate the high selectivity of the two-step labeling

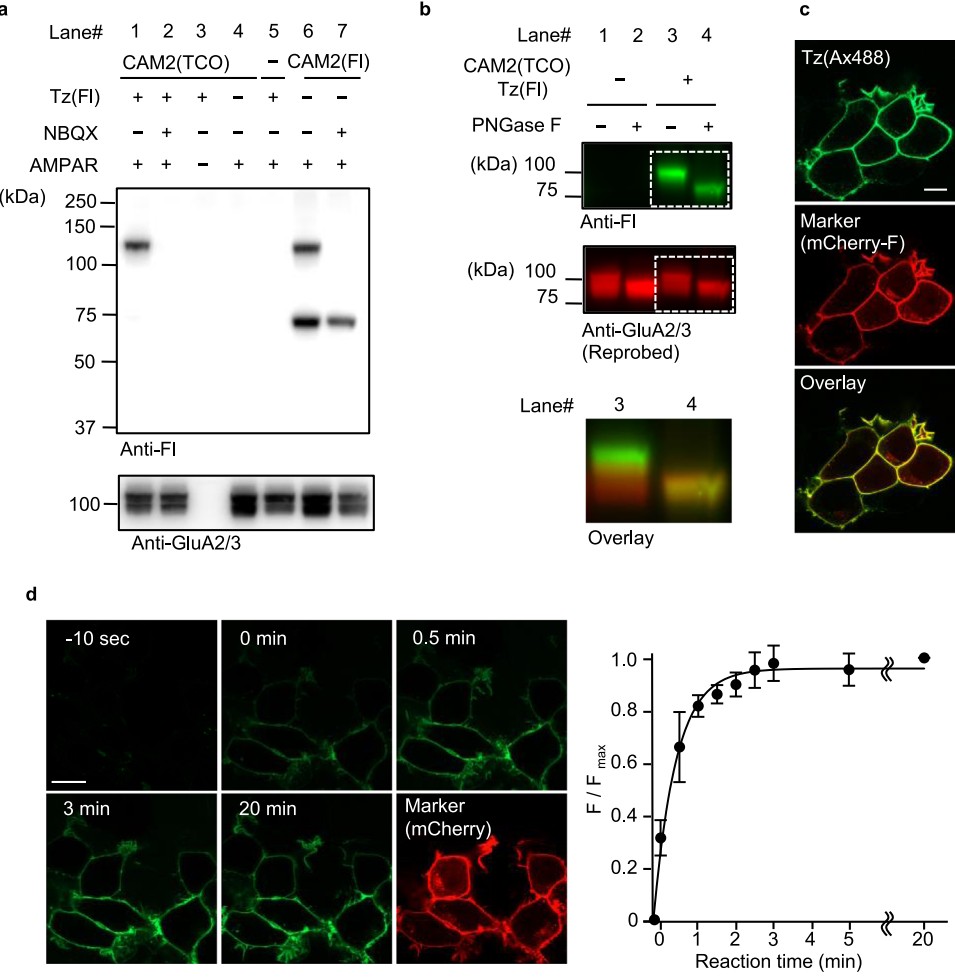

**Fig. 2 The two-step labeling of cell-surface AMPARs ectopically expressed in HEK293T cells. a** Western blotting analyses of HEK293T cells after the two-step labeling. HEK293T cells transfected with GluA2$^{flip}$(Q) (AMPAR(+)) or vector control (AMPAR(−)) were treated with 2 μM CAM2(TCO) for 4 h followed by the addition of 1 μM Tz(Fl) for 5 min, or treated with 2 μM CAM2(Fl) for 4 h in the presence or absence of 50 μM NBQX in culture medium at 37 °C. The cell lysates were analyzed by western blotting using anti-fluorescein or anti-GluA2/3 antibody. Quantification of the band intensity is shown in Supplementary Fig. 3. When CAM2(Fl) is added in serum free medium, strong bands around 70 kDa in lane #6 and #7 disappear (for details see ref. [26]). **b** Effects of PNGase F treatment on the western blotting of labeled AMPAR in HEK293T cells expressing GluA2. Lower image shows the overlay of anti-Fl image and anti-GluA2/3 image for lane #3 and #4. Two-step labeling was conducted as described in (**a**). PNGase F (1000 units/100 μL) was added to the cell lysate. For details, see Methods section. **c** Confocal live imaging of the HEK293T cells labeled with 2 μM CAM2(TCO) and 0.1 μM Tz(Ax488). Labeling was conducted as described in (**a**). mCherry-F was utilized as a transfection marker. Scale bars, 10 μm. **d** Reaction kinetics of tetrazine ligation on live cells by confocal live imaging of the HEK293T cells labeled with 2 μM CAM2(TCO) after addition of 0.3 μM Tz(Ax488) at 37 °C. In left, confocal images are shown. Scale bars, 20 μm. In right, time-course of the fluorescent intensity of Alexa 488 is shown (n = 6 cells). Data are represented as mean ± s.e.m.

technique using CAM2(TCO) compared with the original CAM2(Fl) under cell culture conditions.

For visualizing fluorescently labeled AMPARs on the cell surface, confocal microscopic live imaging was performed after the two-step labeling process under cell culture conditions. Here, Tz(Ax488) was used in the second step of labeling. Alexa 488 has bright fluorescence that is unaffected under endosomal acidic conditions; in contrast, fluorescein has weakened fluorescence under acidic conditions. Thus, Alexa 488 is more suitable to quantify the cellular distribution or trafficking of labeled AMPARs using fluorescent imaging. As shown in Fig. 2c, prominent fluorescence was observed exclusively from the cell surface in cells co-transfected with mCherry-F, a membrane-targeted transfection marker. In contrast, fluorescent signals were not observed in control conditions, such as in CAM2(TCO)-untreated or NBQX-co-treated cells (Supplementary Fig. 4). In the case of direct Alexa 488 labeling using CAM2(Ax488) in the

same cell culture conditions, labeled signals were observed not only from the cell surface but also from the intracellular space (Supplementary Fig. 1c). This suggests that the two-step labeling technique is superior for the fluorescent visualization of cell-surface AMPARs under cell culture conditions. We also determined the reaction kinetics of rapid fluorophore labeling of cell-surface AMPARs with the help of the turn-on fluorescent property of Tz(Ax488) upon the IEDDA reaction (Supplementary Fig. 5). Immediately after adding Tz(Ax488), prominent fluorescent signals were observed from the cells co-transfected with the transfection marker mCherry-F (Fig. 2d), and the fluorescent signals were saturated within 3 min. Thus, cell-surface AMPARs can be labeled by the fluorophore with fast kinetics.

By taking advantage of the high bioorthogonality of the IEDDA reaction, cell-surface AMPARs were successfully labeled with various kinds of chemical probes, ranging from small molecules to middle-sized molecules such as SeTau-647, a squaraine

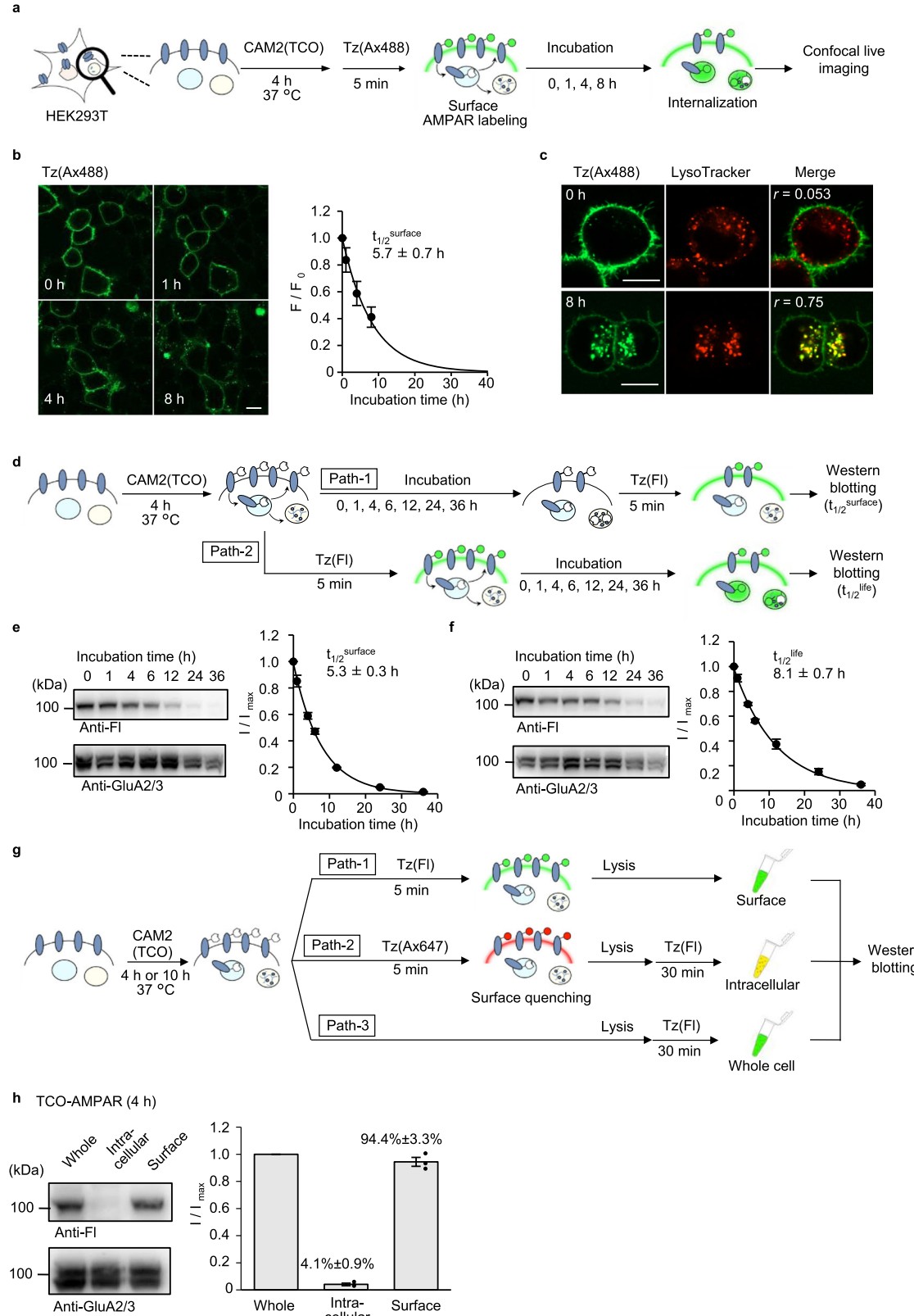

rotaxane dye that has high photostability and a long fluorescence lifetime[39,40] (Supplementary Figs. 2 and 6). This probe flexibility is another feature of the two-step labeling technique that is superior compared with the direct labeling of probes using original CAM2 reagents, where each probe-tethered CAM2 needs to be synthesized (Supplementary Fig. 1b).

With regard to labeling efficacy, quantification of the remaining unlabeled GluA2 fraction showed that $35 \pm 3\%$ of

**Fig. 3 Quantitative analyses of AMPAR trafficking in HEK293T cells using the two-step labeling under the physiological cell culture condition.**
**a** Schematic illustration of trafficking analyses of cell-surface AMPARs by confocal microscopy. **b** Time-lapse confocal imaging of HEK293T cells after two-step labeling using CAM2(TCO) and Tz(Ax488). The HEK293T cells were transfected with GluA2$^{flip}$(Q). In left, confocal images are shown. Scale bars, 10 μm. In right, time-course of the fluorescent intensity from the cell surface is shown ($n = 3$ biological replicates). [CAM2(TCO)] = 2 μM, [Tz(Ax488)] = 0.1 μM. **c** Co-staining of labeled AMPARs with LysoTracker™. The labeled HEK293T cells were incubated with LysoTracker™ Red dnd99 immediately after Tz(Ax488) labeling (upper panel) or subsequent 8 h incubation at 37 °C (lower panel), and confocal live imaging was performed. Pearson's correlation coefficients (r) are shown in the image. Scale bars, 10 μm. **d–f** Determination $t_{1/2}^{surface}$ or $t_{1/2}^{life}$ of AMPARs by western blotting. **d** Schematic illustration of the procedure is shown. In **e** and **f**, $t_{1/2}^{surface}$ and $t_{1/2}^{life}$ are determined, respectively. In left, representative results of western blotting are shown. In right, time-course of the labeled band is shown ($n = 3$ biological replicates). [CAM2(TCO)] = 2 μM, [Tz(Fl)] = 1 μM. **g, h** Determination of TCO-labeled AMPARs on cell-surface, in intracellular area, or in whole-cell by western blotting. In **g**, schematic illustration of the procedure is shown. In **h**, intracellular and surface ratio after CAM2(TCO) labeling for 4 h are determined. In left, representative results of western blotting are shown. In right, band intensities for cell-surface and intracellular labeling were analyzed, both of which were normalized by that for whole-cell labeling ($n = 3$ biological replicates). [CAM2(TCO)] = 2 μM, [Tz(Fl)] = 1 μM. Data are represented as mean ± s.e.m.

surface AMPARs were visualized in the two-step labeling (Supplementary Fig. 7). Moreover, AMPAR function was not visibly affected by the two-step labeling (Supplementary Fig. 8), which is consistent with our previous analyses that showed minimal disturbance of AMPAR ion channel properties by CAM2 labeling[26].

**Analyses of AMPAR trafficking in HEK293T cells**. Once we had a potential labeling method for cell-surface AMPARs under cell culture conditions, we analyzed receptor trafficking using both live imaging and biochemical approaches. First, we analyzed cell-surface AMPAR trafficking in HEK293T cells using confocal live imaging. After incubating the cells with CAM2(TCO) under physiological temperature in culture medium, Tz(Ax488) was added to the culture medium to selectively visualize cell-surface AMPARs and cells were incubated for each period (0–8 h) (Fig. 3a). As shown in Fig. 3b, the labeled fluorescence on the cell surface decreased after incubation at 37 °C. Fluorescent granules were instead observed in the intracellular area, and most of the fluorescent signals were from intracellular granules after 8 h of incubation. The half-time of cell-surface AMPARs ($t_{1/2}^{surface}$), which includes both the remaining and recycled fractions, was calculated to be 5.7 ± 0.7 h from the fluorescent intensity on the cell surface (Fig. 3b). In addition, the intracellular punctate signals merged with a fluorescent lysosome marker (LysoTracker) after 8 h of incubation, suggesting that internalized AMPARs were transported to lysosomes (Fig. 3c). Similar internalization behavior was observed when AMPARs were labeled with different fluorophores using Tz(Ax647) or Tz(ST647) (Supplementary Fig. 9).

Quantitative analyses of the fates of cell-surface AMPARs were examined using biochemical approaches. To quantify the $t_{1/2}^{surface}$ of AMPARs, HEK293T cells were incubated for each period (0–36 h) after treatment with CAM2(TCO) (path-1 in Fig. 3d). Next, Tz(Fl) was added for the selective modification of fluorescein to cell-surface AMPARs. Using western blotting of the cell lysate, the $t_{1/2}^{surface}$ of AMPARs was determined to be 5.3 ± 0.3 h (Fig. 3e), which was similar to the value that was determined using confocal imaging (Fig. 3b). The half-time of the degradation ($t_{1/2}^{life}$) of cell-surface receptors was evaluated by modifying the protocols, where Tz(Fl) was added after CAM2(TCO) labeling (path-2 in Fig. 3d). The cells were then incubated for each period (0–36 h), and the cell lysates were subjected to western blotting. As shown in Fig. 3f, $t_{1/2}^{life}$ was determined to be 8.1 ± 0.7 h, which was slightly longer than the $t_{1/2}^{surface}$ of cell-surface AMPARs ($p < 0.05$). Considering the colocalization of internalized AMPARs and lysosomes (Fig. 3c), the internalized AMPARs are likely decomposed via lysosomal degradation in HEK293T cells.

We next determined the intracellular versus surface percentages of TCO-labeled AMPARs (TCO-AMPARs) in each period,

which can provide valuable information regarding the fate of cell-surface AMPARs. To quantify these percentages, cell-surface AMPARs in HEK293T cells treated with CAM2(TCO) were selectively labeled with fluorescein by adding cell-impermeable Tz (Fl) to the medium under live cell conditions (path-1 in Fig. 3g). To label intracellular TCO-AMPARs, surface TCO-AMPARs were first masked with cell-impermeable Tz(Ax647) (path-2 in Fig. 3g). After lysis of the cells, Tz(Fl) was added to the cell lysate to label intracellular TCO-AMPARs. The whole-cell-labeling fraction, where both cell-surface and intracellular TCO-AMPARs were labeled with fluorescein, was prepared by adding Tz(Fl) after cell lysis (path-3 in Fig. 3g). Prior to these analyses, we first investigated whether the second step reaction using Tz(Fl) proceeds rapidly and/or selectively in cell lysate. Western blotting analyses revealed that the covalent modification of fluorescein was selective to AMPARs in cell lysate (Supplementary Fig. 10), and the reaction was saturated after 15 min when either 0.1 or 0.3 μM Tz(Fl) was added. The intracellular and cell-surface percentages of TCO-AMPARs after 4 h of incubation with CAM2(TCO) were analyzed using this protocol and determined to be 4.1 ± 0.9% and 94.4 ± 3.3%, respectively (Fig. 3h), indicating that intracellular TCO-AMPAR levels were quite low.

**Rapid labeling of endogenous AMPARs in neurons**. We next examined the applicability of the bioorthogonal two-step labeling technique for the rapid modification of cell-surface AMPARs that are endogenously expressed in neurons. Primary cultured neurons from the cerebral cortex were incubated with CAM2(TCO) for 10 h under neuronal culture conditions, and Tz(Fl) was then added for 5 min for cell-surface labeling. Western blotting analyses of the cell lysate showed a single strong band corresponding to the molecular weight of AMPARs (see lane 1 in Fig. 4a). This band was not detected in the co-presence of the competitive ligand NBQX, or in other control conditions (see lanes 2–4 in Fig. 4a). As observed with the AMPARs expressed in HEK293T cells (Fig. 2a), smeared bands were detected using anti-GluA2 antibodies; the anti-Fl band corresponded to the highest band in the smeared anti-GluA2 signals. After the removal of *N*-linked sugars by PNGase treatment, the smeared anti-GluA2 bands converged into a single lower band, which merged with the anti-Fl signal (Supplementary Fig. 11). These results suggest that the highly glycosylated fraction of endogenous AMPARs were selectively labeled with fluorescein by the rapid labeling.

Of the AMPAR subunits (GluA1–4), GluA1, GluA2, and GluA3 are highly expressed in cultured cortical neurons[41]. We next examined the efficacy of our methods for visualizing tetrameric AMPARs by quantifying the remaining unlabeled GluA2 fraction. As shown in Supplementary Fig. 12a, 44 ± 4% of GluA2-containing AMPARs were recognized by the two-step labeling method. Similarly, we calculated that 37 ± 7% of GluA1-

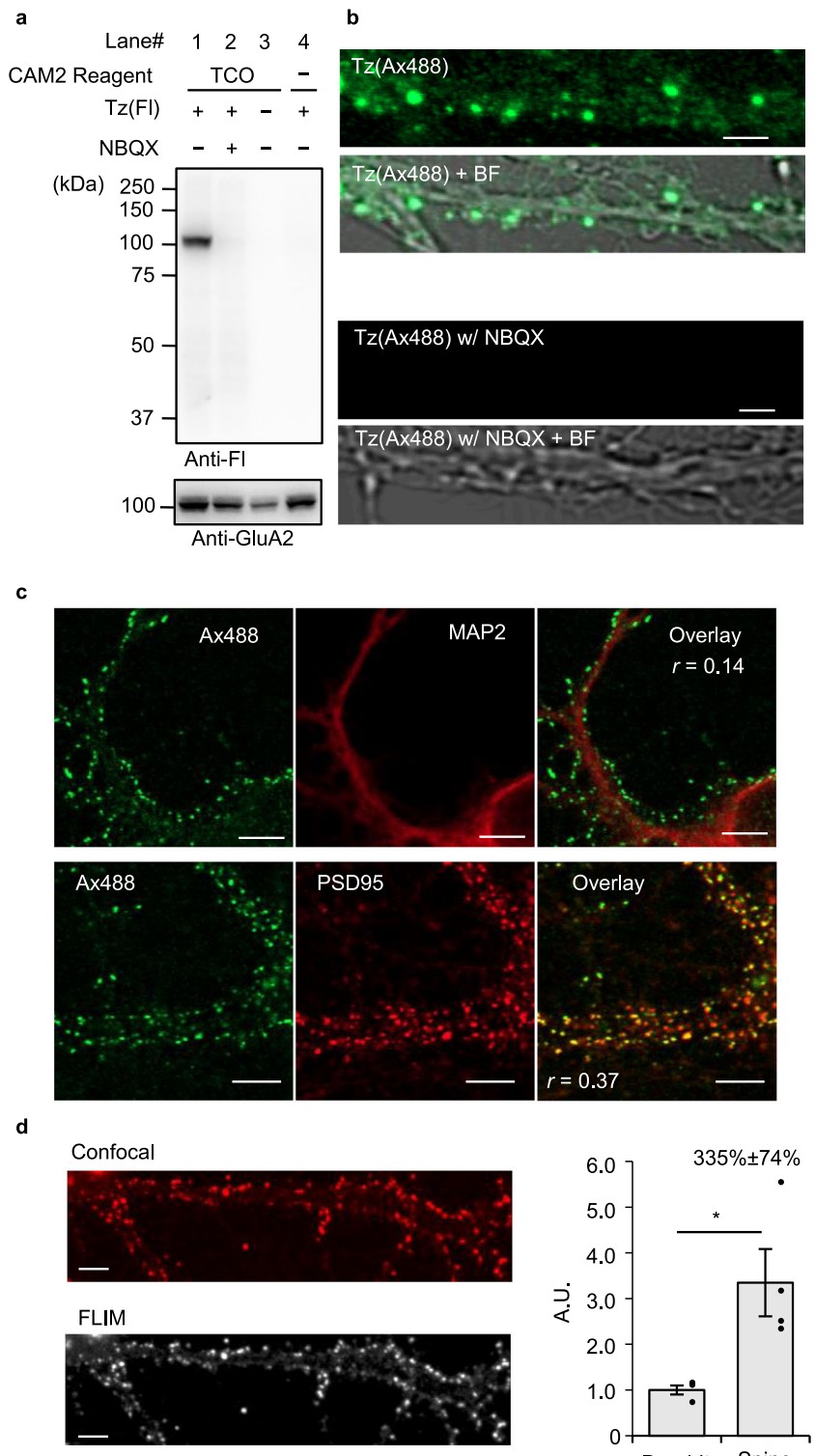

**Fig. 4 The two-step labeling of cell-surface AMPARs endogenously expressed in neurons. a** Western blotting analyses of cortical neurons after the two-step labeling. Primary cultured cortical neurons were treated with 2 μM CAM2(TCO) for 10 h followed by the addition of 1 μM Tz(Fl) for 5 min in the presence or absence of 50 μM NBQX in culture medium at 37 °C. The cell lysates were analyzed by western blotting using anti-fluorescein or anti-GluA2 antibody. **b** Confocal live imaging of the neurons labeled with 2 μM CAM2(TCO) and 0.1 μM Tz(Ax488). Labeling was conducted as described in **a**. Scale bars, 2 μm. **c** Immunostaining of cortical neurons after the two-step labeling. Labeling was conducted as described in **b**. The neurons were fixed, permeabilized, and immunostained using anti-MAP2 (upper) or anti-PSD95 antibody (lower). Scale bars, 5 μm. Pearson's correlation coefficients (*r*) are shown in the image. Whole images are shown in Supplementary Fig. 14a. **d** FLIM imaging and analyses of cell-surface AMPARs in the neurons after the two-step labeling. The neurons were prepared as described in **c**. In left, representative confocal image (upper) and FLIM image (lower) for a lifetime fraction (τ = 2.4 ± 0.1 ns) are shown. Scale bars, 5 μm. In right, FLIM intensities in spine and dendrite were analyzed (n = 4 cells). [CAM2(TCO)] = 2 μM, [Tz(Ax488)] = 0.1 μM. *Significant difference (p < 0.05 by two-sided Student's t-test. p = 0.048). Data are represented as mean ± s.e.m.

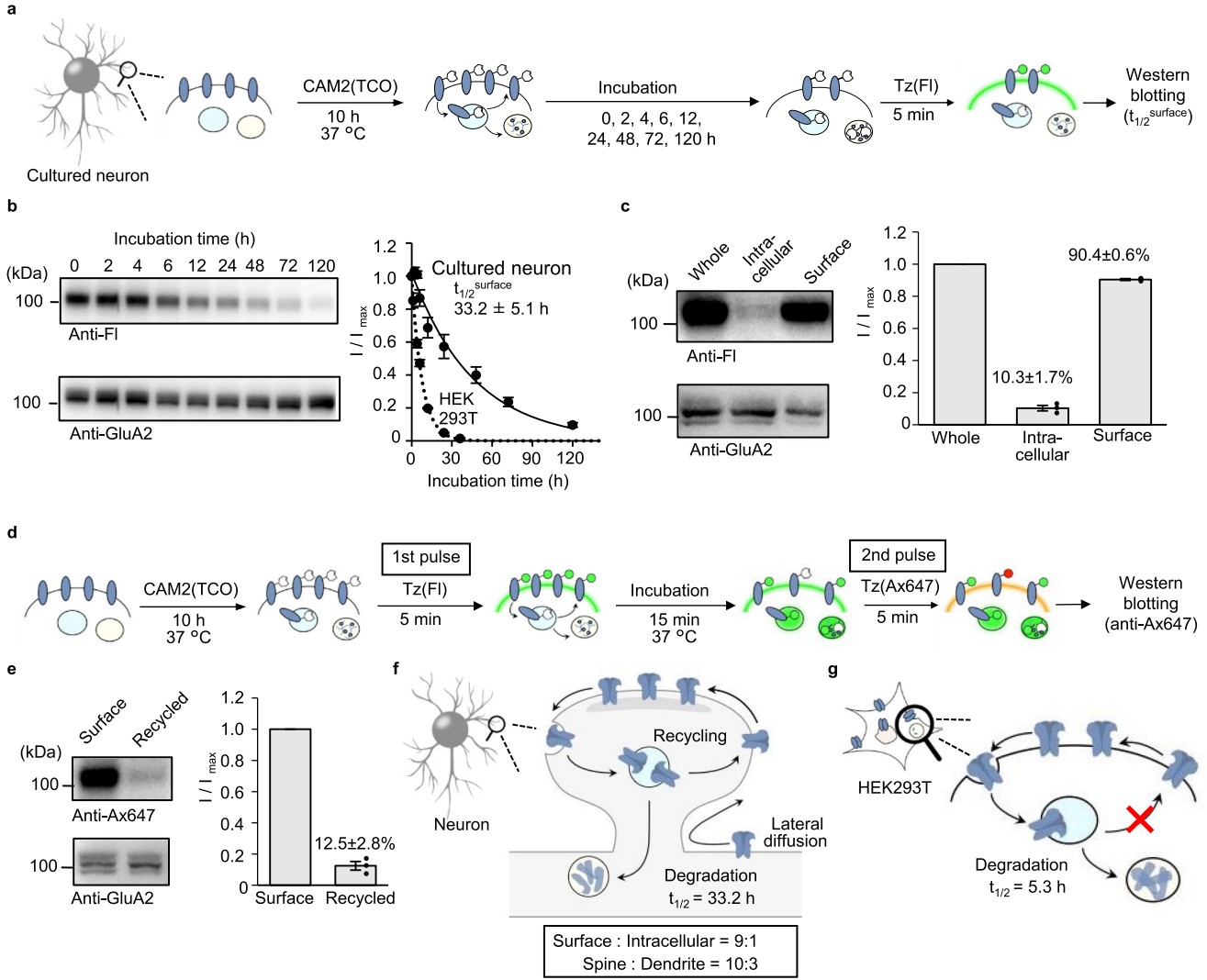

**Fig. 5 Quantitative analyses of AMPAR trafficking in neurons using the two-step labeling under the physiological cell culture condition. a** Schematic illustration of the procedure for determining $t_{1/2}^{surface}$ for AMPARs. **b** Determination of $t_{1/2}^{surface}$ by western blotting. In left, representative results of western blotting are shown. In right, time-course of the labeled band is shown ($n = 3$ biological replicates). [CAM2(TCO)] = 2 μM, [Tz(Fl)] = 1 μM. **c** Determination of intracellular and surface ratio after CAM2(TCO) labeling for 10 h. In left, representative results of western blotting are shown. In right, band intensities for cell-surface and intracellular labeling were analyzed, both of which were normalized by that for whole-cell labeling ($n = 3$ biological replicates). See also Supplementary Fig. 19 for tetrazine ligation in serum containing medium or in cell lysate. [CAM2(TCO)] = 2 μM, [Tz(Fl)] = 1 μM. **d, e** Analyses of recycled AMPARs by pulse-chase-type analyses using the two-step labeling in neurons. In **d**, schematic illustration of the procedure is shown. In **e**, recycled AMPARs were analyzed by western blotting. In left, representative results of western blotting are shown. In right, exocytose AMPARs were quantified, which were normalized by that for surface labeling ($n = 3$ biological replicates). [CAM2(TCO)] = 2 μM, [Tz(Fl) or Tz(Ax647)] = 1 μM. Data are represented as mean ± s.e.m. **f, g** Trafficking and distribution of cell-surface AMPARs quantified by the two-step labeling in neuron (in **f**) and HEK293T cells (in **g**). Data are represented as mean ± s.e.m.

and 43 ± 5% of GluA3-containing AMPARs were recognized. However, considering the heterotetrameric formation of AMPAR subunits, we also needed to examine whether each subunit was covalently labeled with the probe or not. In this context, the immunoprecipitation assay in the denatured condition revealed that GluA2 and GluA3, but not GluA1, were covalently labeled with CAM2(TCO) (Supplementary Fig. 12b). This selectivity is consistent with our previous results[26] in HEK293T cells. With regard to the efficacy of CAM2(TCO) labeling, the time-course of the labeling clearly indicated that chemical labeling occurred more efficiently at 37 °C than in the previous condition at 17 °C (Supplementary Fig. 13a). In addition, the concentration dependency of CAM2(TCO) revealed the EC$_{50}$ value (0.90 ± 0.10 μM) of two-step labeling at 37 °C in neurons (Supplementary Fig. 13b).

Using the two-step labeling technique in primary hippocampal neurons, fluorescently labeled AMPARs were visualized by confocal microscopy. At 5 min after the addition of Tz(Ax488), confocal live imaging showed punctate fluorescent signals from the CAM2(TCO)-treated neurons, and these signals were not observed in neurons co-treated with NBQX (Fig. 4b). To characterize the fluorescent signals in detail, Tz(Ax488)-treated neurons were fixed with paraformaldehyde (PFA) and immunostained with anti-MAP2 or anti-PSD95 antibodies for dendritic or postsynaptic staining, respectively. As shown in Fig. 4c and Supplementary Fig. 14a, labeled Alexa 488 signals were observed alongside the anti-MAP2 signals, and merged well with the anti-PSD95 signals. Considering the short incubation time with Tz (Ax488), the Alexa 488 signal likely corresponds to cell-surface

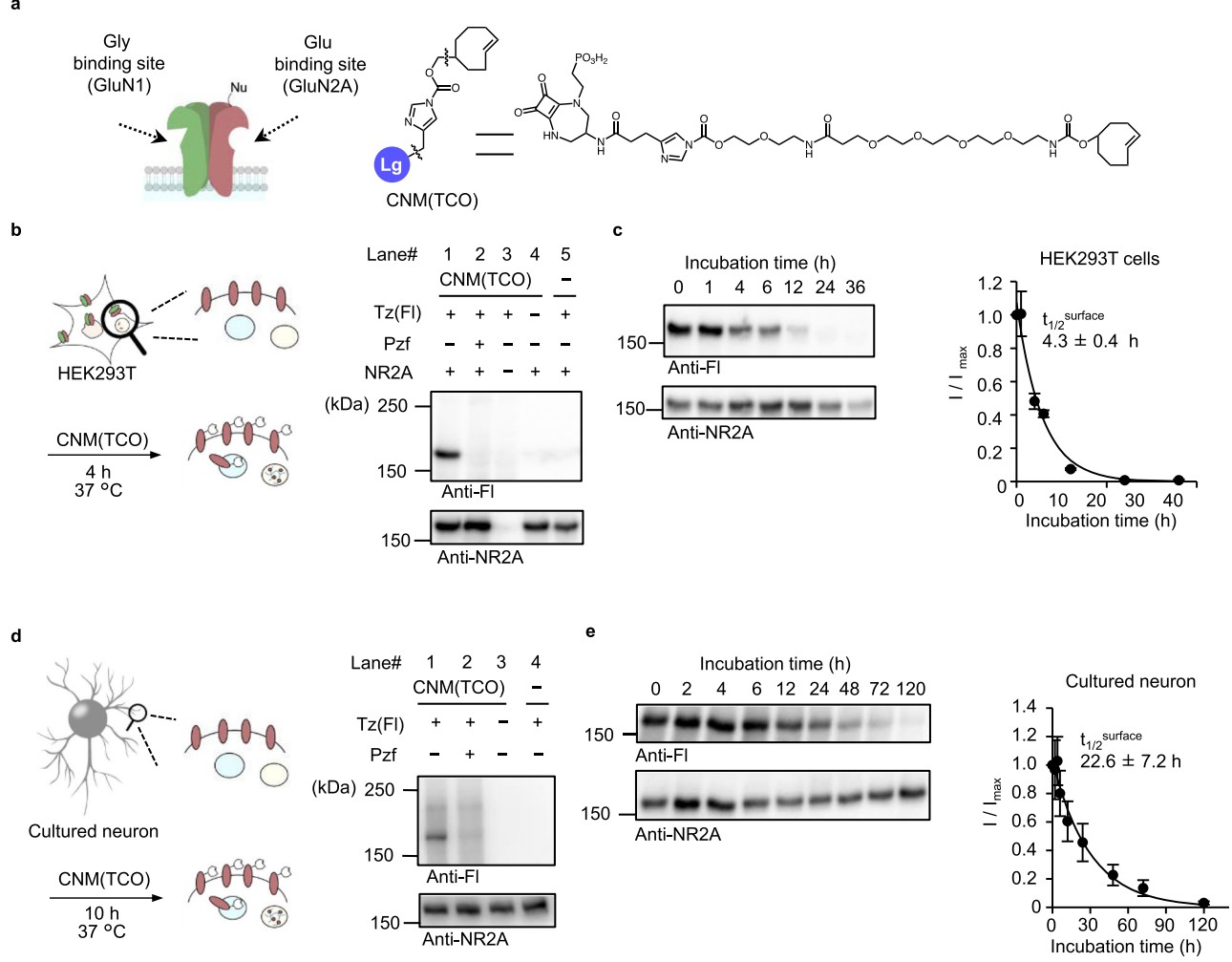

**Fig. 6 The two-step labeling of cell-surface NMDARs in HEK293T cells and neurons. a** Chemical structure of CNM(TCO) for two-step labeling reagent for NMDARs. Gly glycine. **b** Two-step labeling of cell-surface NMDARs ectopically expressed in HEK293T cells. In left, schematic illustration is shown. In right, western blotting analyses after the two-step labeling are shown. HEK293T cells transfected with NR1-1 and NR2A or vector control were treated with 10 μM CNM(TCO) for 4 h followed by the addition of 1 μM Tz(Fl) for 5 min in the presence or absence of 250 μM Pzf in culture medium at 37 °C. The cell lysates were analyzed by western blotting using anti-fluorescein or anti-NR2A antibody. **c** Determination of $t_{1/2}^{surface}$ of NMDARs in HEK293T cells by western blotting. In left, representative results of western blotting are shown. In right, time-course of the labeled band is shown ($n = 3$ biological replicates). **d** Two-step labeling of cell-surface NMDARs endogenously expressed in cultured cortical neurons. In left, schematic illustration is shown. In right, western blotting analyses after the two-step labeling are shown. The cultured cortical neurons were treated with 10 μM CNM(TCO) for 10 h followed by the addition of 1 μM Tz(Fl) for 5 min in the presence or absence of 250 μM Pzf in culture medium at 37 °C. The cell lysates were analyzed by western blotting using anti-fluorescein or anti-NR2A antibody. **e** Determination of $t_{1/2}^{surface}$ of NMDARs in neurons by western blotting. In left, representative results of western blotting are shown. In right, time–course of the labeled band is shown ($n = 3$ biological replicates). Data are represented as mean ± s.e.m.

AMPARs that are endogenously expressed in neurons. In addition, the endogenous AMPARs were successfully visualized using probes such as Alexa 647 or SeTau-647, using Tz(Ax647) or Tz(ST647), respectively (Supplementary Fig. 14b, c).

Next, we quantified the surface distribution of AMPARs in neurons using fluorescence lifetime imaging microscopy (FLIM). In this method, the fluorescence decay curve in each pixel is analyzed by fitting it to a multi-exponential function, and the target fluorescence lifetime (τ) component can be extracted to quantify the fluorescence of interest without any background. Here, we used Tz(ST647) to visualize surface AMPARs because SeTau-647 has a unique fluorescence lifetime and high photostability[39,40]. The typical FLIM image for a lifetime fraction ($\tau = 2.4 \pm 0.1$ ns) corresponding to SeTau-647 is shown in Fig. 4d, which revealed that surface AMPARs in spines are 3.3 times more concentrated than those in dendrites.

**Trafficking analyses of endogenous AMPARs in neurons.** AMPARs are dynamically regulated at synapses, which underlie activity-dependent neuronal plasticity. Molecular biology or biochemical methods, such as the genetic incorporation of fluorescent proteins, surface biotinylation assays, and metabolic incorporation of radioisotopes, have revealed the diffusion dynamics[42], recycling process[5], and half-life[43,44] of AMPARs, respectively. Although powerful, these methods are highly specialized for analyzing each process (see the Discussion for detail). However, we now have a rapid method to selectively label cell-surface AMPARs in neurons under physiological temperature in culture medium. We therefore applied the two-step labeling method to analyze AMPAR trafficking over a long period.

Prior to analyzing the AMPAR trafficking, we examined the influence of the two-step labeling process on the viability of primary cultured neurons by comparing it with our original

CAM2 labeling. In immature neurons (day in vitro [DIV] 4), neither the original CAM2 labeling nor the two-step labeling affected neuronal viability after 24 h of labeling (Supplementary Fig. 15). However, in mature neurons (DIV 12), where iGluRs are highly expressed, viability was decreased after 3 h of labeling with the original CAM2 labeling, and neurons were severely damaged after 24 h of the original labeling. The main reason for the lowered viability may be ascribed to the experimental procedure rather than the CAM2 reagents, because similar neuronal damage was observed when the growth medium was exchanged for serum-free medium. In contrast, in the case of the two-step labeling method, where CAM2(TCO) was directly added into the growth medium at 37 °C, negligible damage was observed even after 24 h of the two-step labeling, indicating that this method is suitable for analyzing AMPARs over long periods in neurons. Besides, we examined the effects of CAM2(TCO) labeling on neuronal function. Neither the surface amount nor the synaptic fraction of AMPARs and NMDARs were affected by CAM2(TCO) labeling for 10 h (Supplementary Fig. 17a, b). Association between GluA2 and the accessary protein, TARPγ8 (ref. [45]) was also unaffected by CAM2(TCO) labeling (Supplementary Fig. 17c). Although the homeostatic phosphorylation of ERK and CREB was not significantly but slightly decreased, phosphorylation levels of AMPAR (GluA1) and NMDAR (GluN1) were not influenced by the CAM2(TCO) labeling (Supplementary Fig. 17d).

We then analyzed the trafficking of endogenous AMPARs in neurons over a long period using the two-step labeling method. As described in the previous section, confocal microscopic imaging was able to clearly visualize cell-surface AMPARs in neurons in the two-step labeling method (Fig. 4c). However, synapses are very narrow (200–500 nm diameter), and it is thus difficult to analyze AMPAR trafficking in detail using optical microscopy. We therefore applied biochemical methods for analyzing the $t_{1/2}^{surface}$ of AMPARs, including remaining and recycled components (Fig. 5a). As shown in Fig. 5b, the $t_{1/2}^{surface}$ was calculated to be 33.2 ± 5.1 h, which was markedly longer than the $t_{1/2}^{surface}$ in HEK293T cells (5.3 ± 0.3 h).

To explore the difference in the $t_{1/2}^{surface}$ values, we focused on the trafficking of internalized AMPARs. We analyzed the intracellular and surface percentages of TCO-AMPARs after 10 h of incubation with CAM2(TCO). The intracellular percentage (10.3 ± 1.7%) was markedly smaller than the surface percentage (90.4 ± 0.6%), and these values were largely similar to those obtained after 10 h of labeling in HEK293T cells (Fig. 5c and Supplementary Fig. 18). We next evaluated the recycling process of internalized AMPARs, as follows. The cultured neurons were labeled with CAM2(TCO) for 10 h in the culture medium at 37 °C, and Tz(Fl) was then added to the culture medium for 5 min as a first pulse to mask the surface TCO-AMPARs (Fig. 5d). The neurons were further incubated for 15 min in the culture medium at 37 °C, and then Tz(Ax647) was added to the culture medium as a second pulse to label the recycled TCO-AMPARs. As shown in Fig. 5e, western blotting using anti-Alexa 647 antibodies clearly detected recycled AMPARs. The percentage was determined to be 12.5 ± 2.8% compared with the surface AMPARs, suggesting that most of internalized AMPARs were recycled during this short period. In contrast, the recycled fraction was almost undetectable in HEK293T cells (Supplementary Fig. 20). These data indicate that AMPARs are constantly recycled via endocytosis and exocytosis in neurons, which may be the molecular basis for the long lifetimes of AMPARs (Fig. 5f, g).

**Two-step labeling and trafficking analyses of NMDARs.** The NMDARs are another essential family of iGluRs, and form

heterotetramers composed of GluN1, GluN2A–D, and GluN3 in neurons. Pharmacological features of NMDARs are different from those of AMPARs. GluN2A–D recognize glutamate, whereas GluN1 and GluN3 recognize glycine or D-serine, as endogenous ligands[4]. Here, we designed a ligand-directed two-step labeling reagent targeted for GluN2A using the selective antagonist perzinfotel (Pzf) as the ligand moiety (Fig. 6a)[46]. This reagent was termed "chemical NMDAR modification (TCO)" (CNM(TCO)).

To confirm the selective labeling of GluN2A, the two-step labeling method was first examined in HEK293T cells transiently co-expressed with GluN1 and GluN2A. Western blotting analyses of cell lysate showed a prominent band around 180 kDa, corresponding to the molecular weight of GluN2A (see lane 1 in Fig. 6b). The 180 kDa band was not detected under the control conditions (see lanes 2–5 in Fig. 6b and Supplementary Fig. 21), suggesting that this band corresponds to labeled GluN2A. The $t_{1/2}^{surface}$ of cell-surface NMDARs was quantitatively analyzed using western blotting by focusing on the labeled GluN2A band. As shown in Fig. 6c, the $t_{1/2}^{surface}$ was calculated to be 4.3 ± 0.4 h, which was slightly shorter than that of AMPARs in HEK293T cells.

Next, we examined the chemical labeling of endogenous NMDARs in cultured neurons using CNM(TCO). As shown in Fig. 6d, western blotting showed a prominent band around 180 kDa, which was negligible in the co-presence of Pzf, indicating the successful labeling of endogenous NMDARs. Similar to AMPAR labeling, the time-course of NMDAR labeling clearly indicated that chemical labeling occurred more efficiently at 37 °C than at 17 °C (Supplementary Fig. 22a). However, the concentration dependency of CNM(TCO) for NMDAR labeling was different from that of CAM2(TCO) for AMPARs. As shown in Supplementary Fig. 22b, the labeled bands were not saturated in the 0–10 μM range, indicating that the affinity of CNM(TCO) was lower than that of CAM2(TCO). Importantly, CNM(TCO) labeled NMDAR with minimal disturbance to neuronal functions, including the constitutive phosphorylation of ERK and CREB (Supplementary Fig. 23). We next analyzed NMDAR trafficking using two-step labeling via biochemical methods in neurons. As shown in Fig. 6e, the $t_{1/2}^{surface}$ of endogenous NMDARs in cultured neurons was 22.6 ± 7.2 h, which was substantially longer than in HEK293T cells. However, the value was shorter than that of endogenous AMPARs in neurons (33.2 ± 5.1 h). Thus, two-step labeling was successfully applied to NMDARs, another important family of iGluRs, by changing the selective ligands. In addition, we revealed that the lifetime of surface NMDARs is shorter than that of AMPARs in neurons.

## Discussion

We described a bioorthogonal two-step labeling method for the selective modification of chemical probes to cell-surface AMPARs. This method can be used under neuronal culture conditions. In our previous method, fluorescein was directly labeled to cell-surface AMPARs using CAM2(Fl) in serum-free medium at 17 °C (ref. [26]). Although the dynamics of cell-surface AMPARs have been successfully tracked over a short period after labeling, the present study revealed that this labeling condition affects the viability of mature neurons after 24 h of labeling. In contrast, the present bioorthogonal two-step labeling method negligibly affected the cell viability of neurons. Our method therefore allows the quantitative analysis of AMPAR trafficking for over 120 h after labeling. The present investigation also revealed that the homeostatic phosphorylation of ERK and CREB was slightly decreased by CAM2(TCO) but not CNM(TCO) treatment. Considering the high affinity of CAM2(TCO) for AMPAR, this influence may be reduced when neurons are treated with low concentration of CAM2(TCO).

Many studies have focused on analyzing AMPAR trafficking in neurons using biochemical methods. To study trafficking over short periods (less than 30 min), surface biotinylation assays, where cleavable (disulfide-linked) biotin is randomly labeled on the cell surface, have conventionally been used. To date, both endocytosis and exocytosis of AMPARs have been investigated in cultured neurons using this method. However, this method is not suitable for analyzing AMPAR trafficking over a long period, because the biotinylation reaction must be conducted in serum-free medium at 4 °C; this affects neuronal viability, as we demonstrated in the present study. The long-term process of AMPAR trafficking, including decomposition, has been previously analyzed using the metabotropic incorporation of a radioisotope [$^{35}$S]-labeled amino acid, or SILAC (stable isotope labeling with amino acids in cell culture), for mass spectrometry analyses[43,44,47]. Although isotope labeling can be conducted under cell culture conditions, these methods are not suitable for analyzing AMPAR trafficking over short periods. In contrast, our bioorthogonal two-step labeling selectively and rapidly modifies target iGluRs under cell culture conditions, and this allows us to analyze receptor trafficking over both short and long periods. Moreover, both biotinylation assays and metabotropic isotope labeling methods require the solubilization of iGluRs by mild detergents for pull-down or immunoprecipitation assays, because cellular proteins are randomly labeled in these methods. This step is problematic for the quantitative analysis of iGluRs, because iGluRs such as NMDARs are mainly localized in the postsynaptic density (PSD), where it is difficult for proteins to be solubilized by mild detergents. This peculiar feature hampers the quantitative analysis of iGluR trafficking by conventional biochemical methods. In contrast, purification steps are not required after labeling in our two-step labeling method. That is, in our method, all labeled proteins in the neurons can be analyzed quantitatively after denaturing in Laemmli sample buffer. This allows for the quantitative analysis of iGluRs contained in the PSD fraction.

The selective visualization of cell-surface iGluRs is essential for analyzing their trafficking and distribution. One potential method involves the use of antibodies that selectively recognize the extracellular regions of iGluRs. However, these antibodies are very limited, and in most cases their selectivity is insufficient. Positron emission tomography imaging would be another candidate for visualizing native AMPARs[48]. However, this method would not be useful for trafficking studies due to its low resolution (1–2 mm). Here, we demonstrated that various kinds of chemical probes could be used to selectively and rapidly label endogenous cell-surface iGluRs in neurons using bioorthogonal two-step labeling. The selective labeling of SeTau-647, a middle-sized molecule with a long fluorescence lifetime and high photostability, allowed the quantitative analyses of cell-surface AMPAR distribution in neurons using FLIM. These analyses revealed a three-fold concentration of cell-surface AMPARs in spines compared with dendrites. As a future direction, cell-surface SeTau-647 labeling would be utilized for single molecule tracking[40] or super resolution imaging of synaptic AMPARs. Moreover, the second step IEDDA reaction can be applied to label polymers or nanoparticles, to allow the distribution of iGluRs to be analyzed in more detail using electron microscopy in the future.

Previous studies using radiolabeling methods indicated that the half-lives of synaptic proteins are 1–2 days[43,44]. Consistent with previous results, our present investigation demonstrated that the half-lives of AMPARs and NMDARs are 33.2 and 22.6 h, respectively, in cultured neurons. However, these values were significantly longer than those obtained in HEK293T cells. A plausible explanation for the differences between neurons and HEK293T cells may involve the formation of the macromolecular protein complexes of iGluRs. In the case of AMPARs, many binding partners such as transmembrane AMPAR regulatory proteins, synapse-associated protein 97 kDa, and glutamate receptor-interacting protein, which are selectively expressed in neurons, control the recycling of AMPARs and/or their stabilization at synapses[45,49,50]. Differences in phosphorylation levels may be another possible explanation. Activity-dependent phosphorylation by CaMKII or PKA contributes to the recycling and cell-surface insertion of AMPARs[51,52]. In addition, ubiquitination or deubiquitination via Nedd4 or USP46, respectively, may be considered[53,54]. In most cases, the contributions of these accessory proteins and post-translational modifications to AMPAR trafficking have been evaluated using genetic approaches, such as overexpression of an AMPAR subunit tagged with pH-sensitive SEP on the N-terminus. However, in some cases, complementary genetic experiments using knock-in or knock-out mice of the target gene have not supported the data[19]. Considering the heterotetrameric formation of AMPARs by the GluA1–4 subunits, and considering that each subunit has inherent roles, conflicting findings may be ascribed to the formation of non-native tetramers by the overexpression of single AMPAR subunits in neurons. In contrast, the present two-step labeling method can be used to visualize native iGluRs under physiological temperature in culture medium. Thus, this method can contribute to our understanding of the physiological and pathophysiological roles of iGluR trafficking in neurons.

## Methods

**Synthesis**. All synthesis procedures and compound characterizations are described in Supplementary Methods.

**General methods for biochemical and biological experiments**. SDS-PAGE and western blotting were carried out using a BIO-RAD Mini-Protean III electrophoresis apparatus. Samples were applied to SDS-PAGE and electrotransferred onto polyvinylidene fluoride membranes (BIO-RAD), followed by blocking with 5% nonfat dry milk in Tris-buffered saline containing 0.05% Tween 20. Primary antibody was indicated in each experimental procedure, and anti-rabbit IgG-HRP conjugate (CST, 7074S, 1:3,000) or anti-mouse IgG-HRP conjugate (CST, 7076S, 1:3,000) was utilized as the secondary antibody. Chemiluminescent signals generated with ECL Prime (GE Healthcare) were detected with a Fusion Solo S imaging system (Vilber Lourmat).

**Animals**. Pregnant ICR mice and pregnant Sprague Dawley rats maintained under specific pathogen-free conditions were purchased from Japan SLC, Inc (Shizuoka, Japan). The animals were housed in a controlled environment (23 ± 1 °C, 12 h light/dark cycle) and had free access to food and water, according to the regulations of the Guidance for Proper Conduct of Animal Experiments by the Ministry of Education, Culture, Sports, Science, and Technology of Japan. All experimental procedures were performed in accordance with the National Institute of Health Guide for the Care and Use of Laboratory Animals, and were approved by the Institutional Animal Use Committees of Kyoto University or Nagoya University.

**Expression of AMPARs or NMDARs in HEK293T cells**. HEK293T cells (ATCC) were cultured in Dulbecco's modified Eagle's medium (DMEM)-GlutaMAX (Invitrogen) supplemented with 10% dialyzed FBS (Invitrogen), penicillin (100 units ml$^{-1}$), and streptomycin (100 μg ml$^{-1}$), and incubated in a 5% CO$_2$ humidified chamber at 37 °C. Cells were transfected with a plasmid encoding rat GluA2 (GluA2$^{flip}$(Q))[26] or the control vector pCAGGS (kindly provided by Dr. H. Niwa from RIKEN) using Lipofectamine 2000 (Invitrogen) according to the manufacturer's instructions, and subjected to labeling experiments after 36–48 h of the transfection. For NMDAR expression, cells were transfected with plasmids encoding rat GluN1-1[55] and rat GluN2A[55], and 30 μM MK-801 (Funakoshi) was added to the culture medium to suppress cell death.

**Two-step labeling of AMPARs or NMDARs in HEK293T cells**. For the first step labeling of AMPARs, HEK293T cells transfected with GluA2 were treated with 2 μM CAM2(TCO) in the absence or presence of 50 μM NBQX in the culture medium at 37 °C for 4 h. For the second step labeling, the culture medium was removed, and 1 μM Tz(Fl) in PBS was added for 5 min at room temperature. To quench excess Tz(Fl), 1 μM TCO-PEG4-COOH in PBS was added.

For western blot analyses of labeled AMPARs, labeled cells were washed three times with PBS, lysed with radio immunoprecipitation assay (RIPA) buffer containing 1% protease inhibitor cocktail (Nacalai tesque), and mixed with 5× Laemmli sample buffer containing 250 mM DTT. Western blotting analyses were performed as described in "General methods for biochemical and biological experiments." The Fl-labeled GluA2 was detected using rabbit anti-fluorescein

antibody (abcam, ab19491, 1:3,000). GluA2 was detected using a rabbit anti-GluA2/3 antibody (Millipore, 07-598, 1:3,000).

In the case of two-step labeling of NMDARs, HEK293T cells transfected with GluN1-1 and GluN2A were treated with 10 μM CNM(TCO) in the absence or presence of 250 μM Pzf in the culture medium at 37 °C for 4 h. The second step labeling and subsequent western blotting were performed as described above. Immunodetection of GluN2A was performed with a rabbit anti-NR2A antibody (Millipore, 07-632, 1:1,000).

CAM2(TCO), CNM(TCO) and Tz-probes were stored in DMSO solution. The stock solutions were kept in deep freezer (−80 °C) to prevent decomposition.

**Enzymatic deglycosylation of AMPARs expressed in HEK293T cells.** GluA2-expressing HEK293T cells were labeled as described in "Two-step labeling of AMPARs or NMDARs in HEK293T cells." The labeled cells were washed three times with PBS and lysed in PBS containing 1% triton X-100, 0.6% SDS, and 1% protease inhibitor cocktail for 30 min at 37 °C. The lysates were diluted (2.0-fold) in sodium phosphate buffer (50 mM, pH 7.5) containing 2% NP40 and 100 mM DTT. PNGase F (New England Biolabs) were used at 1,000 units/100 μl of lysate and incubated overnight at 37 °C. The samples were subjected to western blotting analyses as described in "Two-step labeling of AMPARs or NMDARs in HEK293T cells." In this experiment, after western blotting using anti-Fl antibody, the membrane was stripped with stripping buffer (250 mM glycine (pH = 2.5) and 1% SDS) and reprobed with the anti-GluA2/3 antibody.

**Confocal live cell imaging of labeled AMPARs in HEK293T cells.** HEK293T cells were co-transfected with GluA2$^{flip}$(Q) and mCherry-F[26] as a transfection marker. First step labeling was performed as describe in "Two-step labeling of AMPARs or NMDARs in HEK293T cells." For the second step labeling, after removal of the culture medium, 100 nM Tz(Ax488) was treated for 5 min in HBS (20 mM HEPES, 107 mM NaCl, 6 mM KCl, 2 mM CaCl$_2$, and 1.2 mM MgSO$_4$ at pH 7.4) at room temperature and washed three times with HBS. Confocal live imaging was performed with a confocal microscope (LSM900, Carl Zeiss) equipped with a 63×, numerical aperture (NA) = 1.4 oil-immersion objective. Fluorescence images were acquired by excitation at 405, 488, 561, or 640 nm derived from diode lasers.

For studying the reaction kinetics of tetrazine ligation to cell surface AMPARs, after first step labeling, cells were then incubated with 300 nM Tz(Ax488) at room temperature and imaged at specified time points by confocal microscopy. To quantify the fluorescence intensity of the membrane at each time point, mCherry-F positive cells ($n = 6$) were selected and the average signal intensity of ROIs set on the membrane was calculated by ZEN blue software (Carl Zeiss). After subtracting background fluorescence, the averaged membrane intensity was defined as $F$ at each time point ($F_{MAX}$ was defined as $F$ at 20 min). The membrane intensity was fitted with KaleidaGraph (Synergy software) using following equation (1): $F = a + b (1 - e^{-ct})$.

For trafficking analyses, after first step labeling, 100 nM Tz(Ax488) was added in DMEM-GlutaMAX for 5 min. The cells were washed three times in DMEM-GlutaMAX and incubated for 0, 1, 4, and 8 h in growth medium at 37 °C. Live cell imaging was performed with a confocal microscope. To quantify the fluorescence intensity of the membrane at each time point, mCherry-F positive cells were selected and the average signal intensity of ROIs set on the membrane was calculated by ZEN blue software after subtracting background fluorescence. The averaged membrane intensity was defined as $F$ at each time point. The membrane intensity was fitted with KaleidaGraph using following equation (2): $F = a + b \cdot e^{-ct}$, and the offset value ($a$) was set equal to zero. The $t_{1/2}$ was defined as $t_{1/2} = \ln(2)/c$.

For co-staining with LysoTracker Red DND-99 (Invitrogen), after first step labeling, the cells were treated with 50 nM LysoTracker Red DND-99 for 30 min at 37 °C in culture medium, and then treated with 100 nM Tz(Ax488) in culture medium for 5 min. After washing three times with culture medium or subsequent incubation for 8 h, cells were imaged using a confocal microscope.

**Half-life studies of AMPARs by western blotting in HEK293T cells.** Schematic illustration of the experiments is shown in Fig. 3d. For determining $t_{1/2}$$^{surface}$ (path-1 in Fig. 3d), first step labeling was conducted as described in "Two-step labeling of AMPARs or NMDARs in HEK293T cells." After medium exchange for removal of the labeling reagents, the cells were incubated for 0, 1,4, 6, 12, 24, and 36 h. The cells were treated with 1 μM Tz(Fl) in PBS for 5 min, and excess Tz(Fl) was quenched by addition of 1 μM TCO-PEG4-COOH in PBS.

For determining $t_{1/2}$$^{life}$ (path-2 in Fig. 3d), after first step labeling, cells were washed three times with culture medium and treated with 1 μM Tz(Fl) in culture medium for 5 min. Excess Tz(Fl) was quenched by addition of 1 μM TCO-PEG4-COOH in culture medium. Cells were then incubated for 0, 1,4, 6, 12, 24, and 36 h and washed three times with PBS. Cell lysis and western blotting were performed as described in "Two-step labeling of AMPARs or NMDARs in HEK293T cells."

The target bands were manually selected, and the intensity were calculated with Fusion software (Vilber Lourmat), background intensity was manually subtracted by cutting the minimal intensity in the selected area. The half-life was calculated by curve fitting using KaleidaGraph and following equation (3): $I = a + b \cdot e^{-ct}$, and the offset value ($a$) was set equal to zero. The $t_{1/2}$ was defined as $t_{1/2} = \ln(2)/c$.

**Intracellular and surface ratio of labeled AMPARs in HEK293T cells.** Schematic illustration of the experiments is shown in Fig. 3g. For determining labeled AMPARs on cell surface (path-1 in Fig. 3g), after first step labeling as describe in "Two-step labeling of AMPARs or NMDARs in HEK293T cells," the cells were treated with 1 μM Tz(Fl) for 5 min in PBS at room temperature. To quench excess Tz(Fl), 1 μM TCO-PEG4-COOH in PBS was added and lysed with RIPA buffer containing 1% protease inhibitor cocktail for 30 min at 4 °C.

For determining intracellular labeled AMPARs (path-2 in Fig. 3g), after first step labeling, 1 μM Tz(Ax647) was treated for 5 min for masking of cell-surface TCO-labeled AMPARs. After cell lysis using RIPA buffer, the lysate was reacted with 0.3 μM Tz(Fl) for 30 min at room temperature. Excess Tz(Fl) was quenched by addition of 1 μM TCO-PEG4-COOH in the cell lysate.

For preparing whole-cell-labeling fraction (path-3 in Fig. 3g), after first step labeling, the cells were lysed with RIPA buffer containing 1% protease inhibitor cocktail for 30 min at 4 °C. The lysate was reacted with 0.3 μM Tz(Fl) for 30 min at room temperature. Excess Tz(Fl) was quenched by addition of 1 μM TCO-PEG4-COOH in the cell lysate.

Western blotting was performed as described in "Half-life studies of AMPARs by western blotting in HEK293T cells." The target bands were manually selected, and the intensity were calculated with ImageJ software, background intensity was manually subtracted by selecting a region with no bands from the same lane. In more detail, the band intensity was determined as described below:

$$(\text{target intensity}) - (\text{target area})/(\text{background area}) \times (\text{background intensity})$$

**Preparation of primary cortical neuronal culture.** Twenty-four-well plates (BD Falcon) were coated with poly-D-lysine (Sigma-Aldrich), and washed with sterile dH$_2$O three times. Cerebral cortices from 16-day-old ICR mouse embryos were aseptically dissected and digested with 0.25 w/v% trypsin (Nacalai tesque) for 20 min at 37 °C. The cells were re-suspended in Neurobasal Plus medium supplemented with 10% FBS, penicillin (100 units/ml), and streptomycin (100 μg/ml) and lysed through Cell Strainer (100 μm, Falcon) and centrifuged at 1,000 rpm for 5 min. The cells were re-suspended in Neurobasal Plus medium supplemented with 2% of B-27 Plus Supplement, 1.25 mM GlutaMAX I (Invitrogen), penicillin (100 units/ml), and streptomycin (100 μg/ml) and plated at a density of $2 \times 10^5$ cells on the 24-well plate. The cultures were maintained at 37 °C in a 95% air and 5% CO$_2$ humidified incubator. Culture medium was replaced every 3 or 4 days and the neurons were used at 12–15 DIV.

**Two-step labeling of AMPARs or NMDARs in cultured neurons.** To label endogenous AMPARs, 12 μM CAM2(TCO) in 100 μl of the culture medium with or without 300 μM NBQX was gently added to the cortical neurons cultured in 500 μl medium on 24-well plates to a final concentration of 2 μM CAM2(TCO) and 50 μM NBQX. The cells were incubated for 10 h at 37 °C. For the second step, the culture medium was removed and the cells were treated with 1 μM Tz(Fl) for 5 min in PBS at room temperature. To quench excess Tz(Fl), 1 μM TCO-PEG4-COOH in PBS was added. Western blot analyses of labeled AMPARs were performed as described in "Two-step labeling of AMPARs or NMDARs in HEK293T cells."

To label endogenous NMDARs, 60 μM CNM(TCO) in 100 μl of growth medium with or without 1.5 mM Pzf was gently added to the cortical neurons cultured in 500 μl medium on 24-well plates to a final concentration of 10 μM CNM(TCO) and 50 μM Pzf. The cells were incubated for 10 h at 37 °C. The second step labeling and subsequent western blotting was performed as described above.

**Half-life studies of endogenous AMPARs or NMDARs in cultured neurons.** Schematic illustration of the experiments is shown in Fig. 5a. For determining $t_{1/2}$$^{surface}$, after first step labeling as describe in "Two-step labeling of AMPARs or NMDARs in cultured neurons," the cells were incubated for 0, 2, 4, 6, 12, 24, 48, 72, and 120 h. For the second step, the culture medium was removed and the cells were treated with 1 μM Tz(Fl) for 5 min in PBS at room temperature. To quench excess Tz (Fl), 1 μM TCO-PEG4-COOH in PBS was added and washed three times with PBS. Cell lysis and western blotting were performed as described in "Two-step labeling of AMPARs or NMDARs in HEK293T cells." The immunodetection of GluA2 was conducted with a rabbit anti-GluA2 antibody (abcam, ab20673, 1:3,000). Quantification of the band intensity and calculation of the half-time was calculated described as in "Half-life studies of AMPARs by western blotting in HEK293T cells."

**Intracellular and surface ratio of labeled AMPARs in cultured neurons.** Schematic illustration of the experiments is shown in Fig. 3g. For determining labeled AMPARs on cell surface (path-1 in Fig. 3g), after first step labeling as describe in "Two-step labeling of AMPARs or NMDARs in cultured neurons," the cells were treated with 1 μM Tz(Fl) for 5 min in PBS at room temperature. To quench excess Tz(Fl), 1 μM TCO-PEG4-COOH in PBS was added and lysed with RIPA buffer containing 1% protease inhibitor cocktail for 30 min at 4 °C.

For determining intracellular labeled AMPARs (path-2 in Fig. 3g), after first step labeling, 1 μM Tz(Ax647) was treated for 5 min for masking of cell-surface TCO-labeled AMPARs. After cell lysis using RIPA buffer containing 1% protease inhibitor cocktail for 30 min at 4 °C, the lysate was reacted with 0.3 μM Tz(Fl) for 30 min at room temperature. Excess Tz(Fl) was quenched by addition of 1 μM TCO-PEG4-COOH in the cell lysate.

For preparing whole-cell-labeling fraction (path-3 in Fig. 3g), after first step labeling, the cells were lysed with RIPA buffer containing 1% protease inhibitor cocktail for 30 min at 4 °C. The lysate was reacted with 0.3 μM Tz(Fl) for 30 min at room temperature. Western blotting was performed as described in "Two-step labeling of AMPARs or NMDARs in HEK293T cells." Quantification of the band intensity and calculation of the ratio was conducted as described in "Intracellular and surface ratio of labeled AMPARs in HEK293T cells."

**Quantification of recycled AMPARs in cultured neurons**. The first step labeling was performed as describe in "Two-step labeling of AMPARs or NMDARs in cultured neurons." For the second step, the culture medium was removed and the cells were treated with 1 μM Tz(Fl) for 5 min in the culture medium at 37 °C. To quench excess Tz(Fl), 1 μM TCO-PEG4-COOH in the culture medium was added. After incubation at 37 °C for 15 min, recycled AMPARs were labeled with 1 μM Tz(Ax647) for 5 min in PBS. To quench excess Tz(Ax647), 1 μM TCO-PEG4-COOH in PBS was added. Cell lysis and western blotting were performed as described in "Two-step labeling of AMPARs or NMDARs in HEK293T cells" using anti-Alexa 647 antibody. Quantification of the band intensity and calculation of the ratio was conducted as described in "Intracellular and surface ratio of labeled AMPARs in HEK293T cells."

Anti-Alexa 647 antibody was prepared from the sera of a rabbit immunized with an antigen which was a conjugate of Alexa 647-NHS and KLH (Sigma), and the antibody was affinity-purified using Alexa 647-conjugated agarose. Alexa 647-conjugated agarose was prepared from CarboxyLink Coupling Resin (Thermo Fisher) and Alexa 647 NHS ester (Invitrogen). The anti-sera (1:2,000) or the purified antibody (1:1,000) was used for the western blotting.

**Preparation of primary hippocampal neuronal culture**. Glass bottom dishes (IWAKI) or coverslips (diameter, 13 mm, Matsunami) were coated with poly-D-lysine (Sigma-Aldrich), and washed with sterile $dH_2O$ three times. Hippocampi from 18-day-old Sprague Dawley rat embryos were aseptically dissected and digested with 0.25 w/v% trypsin (Nacalai tesque) for 20 min at 37 °C. The cells were re-suspended in Neurobasal Plus medium supplemented with 10% FBS, penicillin (100 units/ml) and streptomycin (100 μg/ml) and filtered by Cell Strainer (100 μm, Falcon) and centrifuged at 1,000 rpm for 5 min. The cells were re-suspended in Neurobasal Plus medium supplemented with 2% of B-27 Plus Supplement, 1.25 mM GlutaMAX I (Invitrogen), penicillin(100 units/ml), and streptomycin (100 μg/ml) and plated at a density of $2 \times 10^4$ cells on glass coverslips inside 24-well plates (BD Falcon) or glass bottom dishes. Cultures were maintained at 37 °C in a 95% air and 5% $CO_2$ humidified incubator. Culture medium was replaced every 7 days and the neurons were used at 16–18 DIV.

**Live cell imaging of AMPARs in cultured neurons**. To label endogenous AMPARs, 12 μM CAM2(TCO) in 300 μl of growth medium with or without 300 μM NBQX was gently added to the hippocampal neurons cultured in 1.5 ml medium on glass bottom dishes to a final concentration of 2 μM CAM2(TCO) and 50 μM NBQX. After removal of the culture medium, neurons were treated with 100 nM Tz(Ax488) for 5 min in HBS at room temperature and washed three times with HBS. Confocal live imaging was performed with a confocal microscope.

**Immunostaining of cultured neurons after labeling**. Primary cultures of hippocampal neurons were labeled by 2 μM CAM2(TCO) and followed by 100 nM Tz(Ax488) as described above. The cells were fixed with 4% PFA in PBS at room temperature for 30 min and washed three times with PBS. PFA-fixed cells were permeabilized for 15 min with PBS containing 0.1% Triton X-100 at room temperature. The cells were washed three times in PBS and incubated in 10% normal goat serum for 1 h at room temperature. After blocking, the cells were incubated overnight at 4 °C with primary antibodies in PBS containing 1% normal goat serum. The cells were then washed three times with PBS and incubated for 1 h at room temperature with secondary antibodies in PBS containing 1% normal goat serum. The following primary antibodies were used: mouse anti-PSD95 (abcam, ab2723, 1:1,000) or rabbit anti-MAP2 (Millipore, AB5622, 1:1,000). Secondary antibodies were used goat anti-mouse Alexa 647 (abcam, ab150115, 1:1,000) and goat anti-rabbit Alexa633 (Invitrogen, A21070, 1:2,000). Imaging of immunostained hippocampal neurons was performed with a confocal microscope.

**Fluorescence lifetime imaging of AMPARs in cultured hippocampal neurons**. Primary cultures of hippocampal neurons were labeled by 2 μM CAM2(TCO), followed by 100 nM Tz(ST647) and fixed with 4% PFA in PBS. The cells were immunolabeled with PSD95 and MAP2 primary antibodies, and stained with Alexa 488 and Alexa405 secondary antibodies, respectively. Confocal and lifetime imaging of immunostained hippocampal neurons was performed by TCS SP8 FAL-CON (Leica microsystems) equipped with a white light laser and 63×, NA = 1.4 oil-immersion objective. SeTau-647 was imaged using 640 nm exc. (laser power 100, emission collected at 653–700 nm, using 0–12.5 ns time gate).

FLIM images were processed in LAS X 3.5.5 software (Leica microsystems) to fit the lifetime decay curves using an n-exponential reconvolution model with the number of components that $\chi^2$ value is closest to 1. In our data, a three-component fit was utilized. The component of the lifetime corresponding to SeTau-647 ($\tau = 2.4 \pm 0.1$ ns) was used for calculating the intensity of FLIM images. The synaptic or

dendric region was selected ROI on PSD95 or MAP2 signals and the background intensities were subtracted by selecting a region of no cells.

**Statistics and reproducibility**. All graphs were generated using Microsoft Excel. All data are expressed as mean ± s.e.m. We accumulated the data for each condition from at least three independent experiments. We evaluated statistical significance with Student's $t$-test for comparisons between two mean values. A value of $P < 0.05$ was considered significant.

**Reporting summary**. Further information on research design is available in the Nature Research Reporting Summary linked to this article.

## Data availability
The authors declare that the data supporting the findings of this study are available with the paper and its Supplementary information files. The data that support the findings of this study are available from the corresponding author upon reasonable request. Source data are provided with this paper.

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

## Acknowledgements

The authors thank Mr. Hideyuki Yamaguchi (Leica Microsystems) for technical supports in FLIM imaging and Dr. Bronwen Gardner (Edanz Group) for editing a draft of this manuscript. This work was funded by Grants-in-Aid for Scientific Research (KAKENHI) (Grant Number 18J22952 to K.O., 17H06348 to I.H., 16H03290, 19H05778, and 20H02877 to S.K.), Daiichi Sankyo Foundation of Life Science, the Takeda Science Foundation, and the Mochida Memorial Foundation for Medical and Pharmaceutical Research to S.K., and supported by JST CREST (JPMJCR1854) to M.Y. and JST ERATO Grant Number JPMJER1802 to I.H.

## Author contributions

S.K. and I.H. initiated and designed the project. K.O., K.Shiraiwa, K.Soga, T.D., M.T. and K.K. performed synthesis and chemical labeling in HEK293T cells. K.O., K.Shiraiwa, M.Y. and S.K. performed chemical labeling in cultured neurons. K.O. and S.K. wrote the manuscript. All authors discussed and commented on the manuscript.

## Competing interests

The authors declare no competing interests.
