## [Peer Review File · Nature Communications]

REVIEWER COMMENTS

Reviewer #1 (Remarks to the Author):

Ojima et al. present a study on labeling glutamate receptors (AMPA and NMDA) in live neuronal cells with fluorescent markers. The study focuses on combining affinity-based protein labeling with click-chemical modification under physiological conditions. It is based on previous work in which a fluorescent affinity label for AMPA receptors was designed. Click-chemistry provides a fast and efficient bioorthogonal reaction that can be used to link fluorescent dyes to specific protein domains. The labeling approach is of great interest for investigating receptor dynamics including internalization by the cell. The study is well conducted, clearly presented and of great interest to all researchers with an interest for visualizing endogenous glutamate receptors.

When visualizing endogenous receptors both sensitivity and specificity are of interest. While specificity was well studied and discussed in this study, the question remains how efficiently endogenous receptors are recognized. I like to ask the authors, if they could think of any experiment proving that close to all surface exposed receptors are detected in the presented assays. Possibly western blots can be optimized to show and quantify the remaining unlabeled receptor fraction.

Another important question is, if the label influences or even triggers the internalization procedure. I guess there is no experiment that can resolve this in all detail. But, have the authors observed any dependence on the fluorescent dye used in the internalization assays on HEK cells? A discussion on all available evidence that very different labels yield similar half-times for internalization would be helpful.

It has not become clear to me why the intracellular and cell-surface percentages of TCO-AMPA receptors after 4 and 10 h of incubation do not resemble the strong decline of surface signals shown in Fig. 3b, e. Maybe the authors can elaborate some more on this.

When fitting the time-resolved data, the authors applied a fit function with a general offset that is unequal to zero, which clearly influences the determined half-times. How is this motivated and why are the fitted offsets so different for the different time-lapse experiments? Should the offset not be set equal to zero for all experiments?

For most figures the number of independent experiments $n=3$ is mentioned. A description what this refers to (cell preparation, multiple experiments from same cell preparation, repeated label procedures, confocal images) and why such a small number is sufficient is missing.

In the previous publication on CAM it was shown that receptor functionality is unaltered by the labeling procedure. This is probably a complex set of experiments beyond the scope of this manuscript. Still, the authors might want to think of adding such experiments if possible or add a comment on their expectation.

Overall, I very much like the presented work and I recommend publication.

Reviewer #2 (Remarks to the Author):

iGluRs are important in neurotransmission. This manuscript describes a novel approach to label two types of iGluRs (i.e., AMPARs and NMDARs). The manuscript is well-written with a clear structure. The concept described here may also be applied to other protein classes, enabling studying of different endogenous proteins on/in living systems. I would recommend publication of the manuscript after minor corrections.

In the introduction, it would be helpful to expand the description about the possibilities to form different heterotetramers of iGluR, and the implication of this to the currently available approaches. As a chemical biologist with limited knowledge in neurobiology, I would like to know whether heterotetramers are always formed for all three types of iGluRs. How many subtypes are there in each iGluR? What are the known complexes?

Optimization of the labeling conditions. I can't find explanation about the concentration and reaction time of LDAI reagents for AMPARs and NMDARs labeling. Especially, if the previous LDAI

was performed at 17 degrees for 4 h, the currently reported reaction should require less time at 37 degrees (as the reaction rate should be four times as the temperature is 20 degrees higher). I think this is an important point, as currently, the time required for LDAI is longer (e.g., 4-10 h) than the one reported previously.

AMPA(s) labeled in the primary neurons. It would be helpful if the authors could clearly indicate which AMPAR(s) are labeled in the primary neurons with evidences or rationales.

I wonder if the author could provide more details on a few technical aspects:

- Pattern of anti-GluA2/3 seems to be different in Fig 2A. Also, there seems to be two bands in anti-GluA2/3 but just one band in anti-FI. Could the author provide explanations to the difference?
- Can the author calculate the Pearson's correlation coefficient for the merged fluorescence images (e.g., Fig 3c, 4c)?
- Please show anti-GluA2/3 blots for Fig 3e,f,h,i.
- Specificity of CNM(TCO) to GluN2A. I can't find the data demonstrating that CNM(TCO) is not reactive to GluN1 or other NMDARs.
- Include the concentration of CAM2 reagents and tetrazine probes used in all figure captions.
- Statistical analysis. Please include the number of biological- and/or technical replicates used to calculate average and error values. This information is missing in most supplementary figures.
- Please insert the relevant references to Line 23 of page 11.

Reviewer #3 (Remarks to the Author):

In this manuscript, Ojima K. et al. developed an improved approach to label endogenous AMPA receptors with small fluorophores in live cells. Building on their earlier success in labeling AMPARs with LDAI, the authors further incorporated an IEDDA reaction into the process. The advantage of IEDDA reaction is its fast kinetics. This allowed faster labeling under more physiological conditions. The authors applied this method to measure the half-life of AMPARs in HEK 293 cells and cultured neurons. They also developed similar labeling method for NMDARs.

A method to label endogenous AMPAR with small fluorophores under physiological conditions is appealing to scientists in this research field. The approach presented in this study is an incremental advancement from their earlier work (Wakayama S. et al., 2017). While interesting, there are concerns about unsubstantiated claims and flawed analysis as follows. In addition the authors state that their study revealed some unique features of AMPARs, such as a long lifetime and a rapid recycling in neurons

1. The major claim of this manuscript is that the chemical labeling is conducted under physiological conditions. I found this claim is unsubstantiated and overstated.

a. For the first labeling step in neurons, a conjugate of PFQX, a potent AMPAR antagonist, was applied for 10 hr. at a concentration over 10-fold higher than Ki. This may induce homeostatic plasticity in neurons and change the trafficking of AMPARs. The authors should check whether the application of CAM2(TCO) affected the physiology of neurons, including amount of surface and synaptic AMPARs and NMDARs, phosphorylation of AMPAR, NMDAR, CREB and ERK. For the case of NMDAR and Pzf, the same should be examined.

b. Although the first step is conducted under physiological condition (in culture medium at 37 °C), the 2nd step is performed under non-physiological condition (in PBS at room temperature). As we know, solution exchange from culture medium to salt solution acts as a stimulus to neurons. This is especially problematic since all their measurements of half-time, intracellular, surface or recycled percentages of AMPARs assume that surface and intracellular AMPARs are in equilibrium and not affected by the labeling process. Thus, this non-physiological step calls into question of those measurements.

2. The definition of half-time of surface AMPARs is ambiguous. And the determination of this parameter is flawed.

a. HEK293T cells double about every 24-30 hr. The authors did not take this in consideration when

analyzing half-time of surface AMPARs in HEK293T cells with confocal imaging.

b. For WB analysis in both neurons and HEK293T cells, cells were incubated with CAM2(TCO) for 4 or 10 hr., during which time some fraction of surface and intracellular AMPARs were labeled with TCO. During the incubation period of 0-72 hr., TCO-AMPARs are under constant recycling. The measured Tz(FI)-labeled TCO-AMPARs contain two populations: those that remained on the cell surface since time 0; and those that were recycled back to surface. The measurement of half-time of surface AMPARs is complicated by this 2nd population.

3. They mentioned in the discussion that half-lives of synaptic proteins are controversial. Previous work has come to similar conclusions for AMPAR half-lives and rapid recycling so it is not clear how it is controversial. It is also not clear how the new approach resolved the conflicts. Due to the flaws with current analysis, it may add more confusion.

4. The labeling of NMDARs shows much higher background than AMPARs (Figure 6bd). There is labeling even in the absence of TCO (lane 5 in Figure 6b). This indicates this method is case-dependent and limits the general application to other receptors.

5. What's the cell permeability of CAM2(TCO)? What percentage of surface and intracellular receptors in HEK293 cells and neurons were labeled under the conditions used? Does the label affect the channel property, post-translational modifications, or interaction with other associated proteins?

6. When comparing the % recycled in neurons and HEK293T cells, they used two different labeling schemes (Figure 5 and Sp. Figure 11), e.g. the order of Tz(Ax647) and Tz(FI). Due to the possibility that the reaction kinetics and background are different for the two dyes, the same scheme should be used for direct comparison.

7. Do all Tz reagents (Ax488, FI, Ax647) have similar reaction kinetics? The authors should provide these data.

8. What is the meaning of n numbers in the manuscript?

9. Figure 2a: The 70-kDa band is still present although weak. The authors should quantify the intensity.

10. Figure 3b and 5b: the curve does not plateau at 0. Why is that?

11. Figure 3h: the band for "Surface" does not look like 94% of the "Whole". And there are multiple bands for "Intracellular" lane. How was it quantified?

12. It is not clear how the enrichment of surface AMPARs in spines vs. dendrites were quantified (Figure 4d).

13. How neuronal viability was quantified (Sp. Figure 8 and 9)?

14. Sup. Figure 6b and 10b: Why are there two band using anti-GluA2, but single band with anti-FI?

15. p. 16, line 17-20: Need references.

16. p. 17 line 8-9: Need references.

Response to Reviewer 1's comments

General Comments:

Ojima et al. present a study on labeling glutamate receptors (AMPA and NMDA) in live neuronal cells with fluorescent markers. The study focuses on combining affinity-based protein labeling with click-chemical modification under physiological conditions. It is based on previous work in which a fluorescent affinity label for AMPA receptors was designed. Click-chemistry provides a fast and efficient bioorthogonal reaction that can be used to link fluorescent dyes to specific protein domains. The labeling approach is of great interest for investigating receptor dynamics including internalization by the cell. The study is well conducted, clearly presented and of great interest to all researchers with an interest for visualizing endogenous glutamate receptors.

Our response

We would like to thank the reviewer for kind review and for important comments. According to the suggestions and comments, we have carefully amended our manuscript as shown below. All the revisions we made are highlighted in red in the revised manuscript.

Comment 1

When visualizing endogenous receptors both sensitivity and specificity are of interest. While specificity was well studied and discussed in this study, the question remains how efficiently endogenous receptors are recognized. I like to ask the authors, if they could think of any experiment proving that close to all surface exposed receptors are detected in the presented assays. Possibly western blots can be optimized to show and quantify the remaining unlabeled receptor fraction.

Our response

As suggested by this reviewer, we determined how efficiently labeled the native AMPARs in neurons by quantifying the unlabeled AMPARs after the two-step labeling using western blotting.

Prior to the quantification by western blotting, we needed to focus on the existence of several anti-GluA2 bands, which was pointed out by other reviewers (Reviewer #2 (Comment 4) and Reviewer #3 (Comment 14)). In this revision, we revealed that these bands were caused by different glycosylation patterns of GluA2 (see **Supplementary Fig. 11**). In addition, we confirmed that the highest band corresponded to cell-surface GluA2 using surface-biotin-labeling (see **Supplementary Fig. 17a**).

Based on these results, we quantified the remaining unlabeled GluA2 fraction by focusing the highest GluA2 band in neuron. In this experiment, after two-step labeling, FI-labeled AMPARs were immunoprecipitated from the cell lysate using anti-fluorescein antibody. As shown in Supplementary Figure 12a, $44 \pm 4\%$ of GluA2-containing AMPARs were recognized by our present labeling method. Similarly, we calculated that $37 \pm 7\%$ of GluA1- and $43 \pm 5\%$ of GluA3-containing AMPARs were recognized. With regards to GluA4, another AMPAR subunit, western blotting using anti-GluA4 antibody failed to detect GluA4 in our primary cortical culture, which is consistent with previous reports (e.g. ref. 41) showing low level expression of GluA4 in cortical neuronal culture.

However, considering the hetero-tetrameric formation of AMPAR subunits in neurons, we need to confirm whether each subunit is covalently labeled with the probe or not. To address this issue, after two-step labeling using **CAM2(TCO)** and **Tz(FI)**, we examined immunoprecipitation assay by anti-fluorescein antibody using the cell lysate in the denatured condition under which tetrameric formation of AMPARs is collapsed. As shown in Supplementary Figure 12b, the immunoprecipitation assay revealed that **CAM2(TCO)** was covalently labeled to GluA2 or GluA3 but not to GluA1. This selectivity is consistent with our previous results showing covalent labeling to GluA2, GluA3 or GluA4 among AMPAR subunits (GluA1–4) in HEK293T cells (ref. 26).

These results indicate that all of AMPARs are not chemically labeled in our method. However, we think that it is not problematic in the analyses of AMPARs in neurons because sufficient fluorescent intensity or strong band intensity is detected in confocal imaging or western blotting, respectively.

Modification in the main text

In page 11, line 13–: **As observed with the AMPARs expressed in HEK293T cells (Fig. 2a), smeared bands were detected using anti-GluA2 antibodies; the anti-FI band corresponded to the highest band in the smeared anti-GluA2 signals. After the removal of N-linked sugars by PNGase treatment, the smeared anti-GluA2 bands converged into a single lower band, which merged with the anti-FI signal (Supplementary Fig. 11).** These results suggest that the highly glycosylated fraction of endogenous AMPARs were selectively labeled with fluorescein by the rapid labeling.

Of the AMPAR subunits (GluA1–4), GluA1, GluA2 and GluA3 are highly expressed in cultured cortical neurons⁴¹. We next examined the efficacy of our methods for visualizing tetrameric AMPARs by quantifying the remaining unlabeled GluA2 fraction. As shown in Supplementary Figure 12a, $44 \pm 4\%$ of GluA2-containing AMPARs were recognized by the two-step labeling method. Similarly, we calculated that $37 \pm 7\%$ of GluA1- and $43 \pm 5\%$ of GluA3-containing AMPARs were recognized. However, considering the heterotetrameric formation of AMPAR subunits, we also needed to examine whether each subunit was covalently labeled with the probe or not. In this context, the immunoprecipitation assay in the denatured condition revealed that GluA2 and GluA3, but not GluA1, were covalently labeled with **CAM2(TCO)** (**Supplementary Fig. 12b**). This selectivity is consistent with our previous results²⁶ in HEK293T cells.

Modification in Figures or Supplementary information

- Supplementary Figure 11: Deglycosylation assay of GluA2 in neuron was newly added.
- Supplementary Figure 12: Labeling efficacy and subunit selectivity of the labeling were newly added.
- Supplementary Figure 17a: Surface-biotin-labeling of GluA2 in neuron was newly added.

Comment 2

Another important question is, if the label influences or even triggers the internalization procedure. I guess there is no experiment that can resolve this in all detail. But, have the authors observed any dependence on the fluorescent dye used in the internalization assays on HEK cells? A discussion on all available evidence that very different labels yield similar half-times for internalization would be helpful.

Our response

We evaluated influence of the fluorescent dye in the internalization assay in HEK293T cells. As suggested by this review, we changed labeled fluorophore from Alexa488 to structurally different fluorophores such as Alexa 647 and SeTau 647. As shown in Supplementary Figure 9, internalization was not prominently affected by changing the labeled fluorophore utilized in this paper.

Modification in the main text

In page 9, line 23–: **Similar internalization behavior was observed when AMPARs were labeled with different fluorophores using Tz(Ax647) or Tz(ST647) (Supplementary Fig. 9).**

Modification in Figures or Supplementary information

- Supplementary Figure 9: Internalization assay of AMPAR labeled with Alexa 647 or SeTau 647 in HEK293T cells were newly added.

Comment 3

It has not become clear to me why the intracellular and cell-surface percentages of TCO-AMPARs after 4 and 10 h of incubation do not resemble the strong decline of surface signals shown in Fig. 3b, e. Maybe the authors can elaborate some more on this.

Our response

This discrepancy is ascribed to the difference of experimental procedures. For calculating surface and intracellular AMPARs in Figure 3h, **CAM2(TCO)** was added to the culture medium of HEK293T cells transiently expressed with GluA2 for 4 h (**Fig. 3h**) and 10 h (in original **Fig. 3i** (current **Supplementary Fig. 18**)). In these experiments, the cells were subjected to 2nd step labeling using Tz-probes (**Tz(FI)** or **Tz(Ax647)**) immediately after the TCO-labeling to AMPARs, to quantify the surface, intracellular or whole cell fraction of labeled AMPARs. In contrast, in the case of determining half-time of surface AMPAR ($t_{1/2}^{surface}$) in Figure 3b and 3e, remaining labeling reagents were removed by medium exchange after **CAM2(TCO)** labeling for 4 h. Then, the cells were incubated for each period (0–36 h) to track the internalization or decomposition. As pointed out by this reviewer, the original arrangement of these figures may confuse readers. To improve this issue, we moved original Figure 3i (**CAM2(TCO)** labeling for 10 h) to Supplementary Figure 18 in this revision. We believe that this rearrangement would be helpful for readers because this data (**CAM2(TCO)** labeling for 10 h in HEK293T cells) is a control experiment of Figure 5c data (**CAM2(TCO)** labeling for 10 h in neurons). Thank you for your helpful comment.

With regards to high surface ratio even after **CAM2(TCO)** labeling for 10 h, one plausible explanation is that decomposition of labeled AMPARs coincides with labeling of newly synthesized AMPARs in HEK293T cells. Actually, as shown in the attached Figure A, the labeling yields were not largely different between 4 h-labeled cells and 10-h labeled cells in HEK293T cells, in spite of relatively short half-life ($t_{1/2}^{life} = 8.1 \pm 0.7$ h) of AMPARs. This result indicates that synthesis of AMPARs simultaneously occurs with decomposition in HEK293T cells transiently expressed with AMPARs.

Figure A. Two-step labeling of cell-surface AMPARs in HEK293T cells. The cells were incubated with 2 μM CAM2(TCO) for 4h or 10h followed by the addition of 1 μM Tz(FI) for 5min at 37 °C. Western blots using anti-FI or anti-GluA2/3 of the cell lysate was shown.

Modification in Figures or Supplementary information

- Supplementary Figure 18: Original Figure 3i is moved to Supplementary Figure 18.

Comment 4

When fitting the time-resolved data, the authors applied a fit function with a general offset that is unequal to zero, which clearly influences the determined half-times. How is this motivated and why are the fitted offsets so different for the different time-lapse experiments? Should the offset not be set equal to zero for all experiments?

Our response

In the original manuscript, offset value of the fit function was not set equal to zero. The reason was that our original manuscript lacked the data where most of labeled AMPARs were decomposed or internalized. In this revision, we re-evaluated the internalization or degradation analyses by incubating the cells for longer periods (0 – 36 h in HEK cells or 0 – 120 h in neurons). As a result, offset value could be set equal to zero in new data. Importantly, although the newly obtained $t_{1/2}$ values were slightly different from original values, the tendency and our claim were not changed at all.

Modification in the main text

In page 23, line 12–: The membrane intensity was fitted with KaleidaGraph using following equation : $F = a + b \cdot e^{-ct}$, and the offset value (a) was set equal to zero. The $t_{1/2}$ was defined as $t_{1/2} = \ln(2)/c$.

In page 24, line 9–: The half-life was calculated by curve fitting using KaleidaGraph and following equation: $I = a + b \cdot e^{-ct}$, and the offset value (a) was set equal to zero. The $t_{1/2}$ was defined as $t_{1/2} = \ln(2)/c$.

Modification in Figures or Supplementary information

- Figure 3b: The offset value (a) was set equal to zero using the original data.
- Figure 3e and 3f: The data was revised by changing the incubation time from 0–24 h to 0–36 h.
- Figure 5b: The data was revised by changing the incubation time from 0–72 h to 0–120 h.
- Figure 6c: The data was revised by changing the incubation time from 0–24 h to 0–36 h.
- Figure 6e: The data was revised by changing the incubation time from 0–72 h to 0–120 h.

Comment 5

For most figures the number of independent experiments $n=3$ is mentioned. A description what this refers to (cell preparation, multiple experiments from same cell preparation, repeated label procedures, confocal images) and why such a small number is sufficient is missing.

Our response

As suggested by this reviewer, we added description about the number of samples in each figure legend. Most experiments were conducted triplicate or more in biologically distinct samples, which is clearly described in the revised manuscript. In addition, both cell numbers and biological replicates were described in Supplementary Figure 8 and 15 legend. We believe that biological triplicate is sufficient to analyze our biochemical data.

Comment 6

In the previous publication on CAM it was shown that receptor functionality is unaltered by the labeling procedure. This is probably a complex set of experiments beyond the scope of this manuscript. Still, the authors might want to think of adding such experiments if possible or add a comment on their expectation.

Our response

As commented by this reviewer, we previously confirmed that CAM2(FI) labeling did not affect the concentration dependency of glutamate or ion-channel kinetics of AMPARs by electrophysiological assay (ref. 26: Wakayama *et al. Nat. Commun.* 2017). With regards to CAM2(TCO) utilized in this paper, the linker length between ligand group and reactive acyl imidazole group is completely same with that of CAM2(FI). Thus, it is expected that two-step labeling using CAM2(TCO) has little effect on AMPAR function. In addition, in this revision, we evaluated effect of two-step labeling using CAM2(TCO)/Tz(FI) pair or CAM2(TCO)/Tz(Ax488) pair on the AMPAR response. As shown in Supplementary Figure 8, Ca²⁺ imaging indicated that two-step labeling did not prominently affect the concentration dependency of glutamate in HEK293T cells transiently expressed with GluA2. This indicate that we can label fluorophores to AMPAR with negligible disturbance of the receptor function using the two-step labeling.

Modification in the main text

In page 9, line 5–: **Moreover, AMPAR function was not visibly affected by the two-step labeling (Supplementary Fig. 8), which is consistent with our previous analyses that showed minimal disturbance of AMPAR ion channel properties by CAM2 labeling²⁶.**

Modification in Figures or Supplementary information

- Supplementary Figure 8: Effects of the two-step labeling to AMPAR function was newly added.

Response to Reviewer 2's comments

General Comments:

iGluRs are important in neurotransmission. This manuscript describes a novel approach to label two types of iGluRs (i.e., AMPARs and NMDARs). The manuscript is well-written with a clear structure. The concept described here may also be applied to other protein classes, enabling studying of different endogenous proteins on/in living systems. I would recommend publication of the manuscript after minor corrections.

Our response

We would like to thank the reviewer for kind review and for important comments. According to the suggestions and comments, we have carefully amended our manuscript as shown below. All the revisions we made are highlighted in red in the revised manuscript.

Comment 1

In the introduction, it would be helpful to expand the description about the possibilities to form different heterotetramers of iGluR, and the implication of this to the currently available approaches. As a chemical biologist with limited knowledge in neurobiology, I would like to know whether heterotetramers are always formed for all three types of iGluRs. How many subtypes are there in each iGluR? What are the known complexes?

Our response

Thank you for your helpful comment. We added the description about subtypes and their heterotetrameric formation of iGluRs in the introduction part.

Modification in the main text

In page 3, line 2–: In the central nervous system, ionotropic glutamate receptors (iGluRs) mediate fast excitatory neurotransmission. iGluRs are categorized into **distinct classes based on their pharmacology and structural homology, including** the α -amino-3-hydroxy-5-methyl-4-isoxazole-propionate (AMPA) receptor (**GluA1–4**), kainate receptor (**GluK1–5**), *N*-methyl-D-aspartate (NMDA) receptor (**GluN1, GluN2A–D, GluN3A–B**) and δ receptors (**GluD1–2**)¹. **Glutamate receptors assemble as tetramers, and functional receptors are formed exclusively by the assembly of subunits within the same functional receptor class.**

AMPA receptors (AMPARs), which are mainly permeable to monovalent cations (Na^+ and K^+), mediate the majority of excitatory synaptic transmission. **AMPARs can form homotetramers or heterotetramers, and subunit compositions are dependent on brain regions. In hippocampal CA1 neurons, the majority of AMPARs are made up of GluA1/A2 and GluA2/A3 subunit combinations, with a small contribution of GluA1 homomers**^{2,3}. Recent studies have revealed that AMPARs are constitutively cycled in and out of the postsynaptic membrane through endocytosis and exocytosis. The precise regulation of this process is critical for synaptic plasticity, which is the basis of learning, memory, and

development in neural circuits^{2,3}. Although AMPARs and kainate receptors are activated by glutamate binding, NMDA receptors (NMDARs), which have high permeability to Ca²⁺, require depolarization as well as agonist binding for their activation. **Functional NMDARs require the assembly of two GluN1 subunits together with either two GluN2 subunits, or a combination of GluN2 and GluN3 subunits¹.**

Comment 2

Optimization of the labeling conditions. I can't find explanation about the concentration and reaction time of LDAI reagents for AMPARs and NMDARs labeling. Especially, if the previous LDAI was performed at 17 degrees for 4 h, the currently reported reaction should require less time at 37 degrees (as the reaction rate should be four times as the temperature is 20 degrees higher). I think this is an important point, as currently, the time required for LDAI is longer (e.g., 4-10 h) than the one reported previously.

Our response

As suggested by this reviewer, we evaluated the time-course of the AMPAR or NMDAR labeling at 17 °C or 37 °C in neurons (**Supplementary Fig. 13 and 22**). As shown in Supplementary Figure 13a, reaction rate of **CAM2(TCO)** labeling to AMPAR at 37 °C was higher than that at 17 °C. This indicates that we can shorten the labeling period at 37 °C as suggested by this reviewer. With regards to **CNM(TCO)** labeling for NMDARs, similar temperature dependency of the reaction rate was observed (**Supplementary Fig. 22a**).

We also examined concentration dependency of these LDAI reagents. As shown in Supplementary Fig. 13b, the EC₅₀ value of the AMPAR labeling using **CAM2(TCO)** was determined as 0.90 ± 0.10 μM. However, in the case of NMDAR labeling using **CNM(TCO)**, saturation of the labeled band was not observed in the concentration range ([**CNM(TCO)**] = 0 – 10 μM) (**Supplementary Fig. 22b**). These indicate that the affinity of **CNM(TCO)** to NMDARs is not so high compared with that of **CAM2(TCO)** to AMPARs.

Modification in the main text

In page 12, line 4–: **With regards to the efficacy of CAM2(TCO) labeling, the time-course of the labeling clearly indicated that chemical labeling occurred more efficiently at 37 °C than in the previous condition at 17 °C (Supplementary Fig. 13a). In addition, the concentration dependency of CAM2(TCO) revealed the EC₅₀ value (0.90 ± 0.10 μM) of two-step labeling at 37 °C in neurons (Supplementary Fig. 13b).**

In page 16, line 4–: **Similar to AMPAR labeling, the time-course of NMDAR labeling clearly indicated that chemical labeling occurred more efficiently at 37 °C than at 17 °C (Supplementary Fig. 22a). However, the concentration dependency of CNM(TCO) for NMDAR labeling was different from that of CAM2(TCO) for AMPARs. As shown in Supplementary Figure 22b, the labeled bands were not saturated in the 0–10 μM range, indicating that the affinity of CNM(TCO) was lower than that of CAM2(TCO).**

Modification in Figures or Supplementary information

- Supplementary Figure 13: Time course and concentration dependency of **CAM2(TCO)** labeling to AMPARs in neurons was newly added.
- Supplementary Figure 22: Time course and concentration dependency of **CNM(TCO)** labeling to NMDARs in neurons was newly added.

Comment 3

AMPAR(s) labeled in the primary neurons. It would be helpful if the authors could clearly indicate which AMPAR(s) are labeled in the primary neurons with evidences or rationales.

Our response

As suggested by this reviewer, we determined subunit selectivity in primary neurons. First, we determined how efficiently labeled the native AMPARs in neurons by quantifying the unlabeled AMPARs after the two-step labeling using western blotting.

Prior to the quantification by western blotting, we needed to focus on the existence of several anti-GluA2 bands, which was pointed out by this reviewer (Comment 4) and reviewer #3 (Comment 14). In this revision, we revealed that these bands were caused by different glycosylation patterns of GluA2 in neurons (see **Supplementary Fig. 11**). In addition, we confirmed that the highest band corresponds to cell-surface GluA2 using surface-biotin-labeling (see **Supplementary Fig. 17a**).

Based on these results, we quantified the remaining unlabeled GluA2 fraction by focusing the highest GluA2 band in neuron. In this experiment, after two-step labeling, FI-labeled AMPARs were immunoprecipitated from the cell lysate using anti-fluorescein antibody. As shown in Supplementary Figure 12a, $44 \pm 4\%$ of GluA2-containing AMPARs were recognized by our present labeling method. Similarly, we calculated that $37 \pm 7\%$ of GluA1- and $43 \pm 5\%$ of GluA3-containing AMPARs were recognized. With regards to GluA4, another AMPAR subunit, western blotting using anti-GluA4 antibody failed to detect GluA4 in our primary cortical culture, which is consistent with previous reports (e.g. ref. 41) showing low level expression of GluA4 in cortical neuronal culture.

However, considering the heterotetrameric formation of AMPAR subunits in neurons, we need to confirm whether each subunit is covalently labeled with the probe or not. To address this issue, after two-step labeling using **CAM2(TCO)** and **Tz(FI)**, we examined immunoprecipitation assay by anti-fluorescein antibody using the cell lysate in the denatured condition under which tetrameric formation of AMPARs is collapsed. As shown in Supplementary Figure 12b, the immunoprecipitation assay revealed that **CAM2(TCO)** was covalently labeled to GluA2 or GluA3 but not to GluA1. This selectivity is consistent with our previous results showing covalent labeling to GluA2, GluA3 or GluA4 among AMPAR subunits (GluA1–4) in HEK293T cells (ref. 26). These results indicate that GluA2 and GluA3 are covalently labeled with **CAM2(TCO)**, and that heteromers containing GluA1 are visualized in primary cortical neurons in our method.

Modification in the main text

In page 11, line 13–: **As observed with the AMPARs expressed in HEK293T cells (Fig. 2a), smeared bands were detected using anti-GluA2 antibodies; the anti-FI band corresponded to the highest band in**

the smeared anti-GluA2 signals. After the removal of *N*-linked sugars by PNGase treatment, the smeared anti-GluA2 bands converged into a single lower band, which merged with the anti-FI signal (**Supplementary Fig. 11**). These results suggest that the highly glycosylated fraction of endogenous AMPARs were selectively labeled with fluorescein by the rapid labeling.

Of the AMPAR subunits (GluA1–4), GluA1, GluA2 and GluA3 are highly expressed in cultured cortical neurons⁴¹. We next examined the efficacy of our methods for visualizing tetrameric AMPARs by quantifying the remaining unlabeled GluA2 fraction. As shown in Supplementary Figure 12a, $44 \pm 4\%$ of GluA2-containing AMPARs were recognized by the two-step labeling method. Similarly, we calculated that $37 \pm 7\%$ of GluA1- and $43 \pm 5\%$ of GluA3-containing AMPARs were recognized. However, considering the heterotetrameric formation of AMPAR subunits, we also needed to examine whether each subunit was covalently labeled with the probe or not. In this context, the immunoprecipitation assay in the denatured condition revealed that GluA2 and GluA3, but not GluA1, were covalently labeled with CAM2(TCO) (**Supplementary Fig. 12b**). This selectivity is consistent with our previous results²⁶ in HEK293T cells.

Modification in Figures or Supplementary information

- Supplementary Figure 11: Deglycosylation assay of GluA2 in neuron was newly added.
- Supplementary Figure 12: Labeling efficacy and subunit selectivity of the labeling were newly added.
- Supplementary Figure 17a: Surface-biotin-labeling of GluA2 in neuron was newly added.

Comment 4

I wonder if the author could provide more details on a few technical aspects:

Comment 4-1

Pattern of anti-GluA2/3 seems to be different in Fig 2A. Also, there seems to be two bands in anti-GluA2/3 but just one band in anti-FI. Could the author provide explanations to the difference?

Our response

Previous studies have revealed that GluA2 undergo the posttranslational modification by *N*-linked glycosylation (e.g. ref. 38). Thus, a plausible explanation of the difference is that these bands correspond to the different glycosylation pattern of GluA2. To examine this possibility, we treated the cell lysate with peptide-*N*-glycosidase F (PNGase F) for removal of all *N*-linked sugars. As shown in Figure 2b, after treatment with PNGase F, western blotting using anti-GluA2/3 showed single band which shifted to the lower molecular weight direction. This indicates that these bands in anti-GluA2/3 correspond to a different glycoform pattern of GluA2.

With regards to detection of one band by anti-fluorescein antibody, molecular weight of the labeled band corresponds to the highest band in anti-GluA2/3 signals (see **Fig. 2b**). After treatment of PNGase F to these samples, molecular weight of the anti-FI band shifted toward lower molecular weight, and the molecular weight was comparable to that of the deglycosylated GluA2 band (**Fig. 2b**). These indicate that the highest band in anti-GluA2/3 is selectively labeled by our methods. Besides, similar

results were obtained for endogenous AMPARs in neurons (**Supplementary Fig. 11**). Surface biotinylation assay further revealed that the highest band corresponded to cell-surface GluA2 in neuron (**Supplementary Fig. 17**). These results support that the selective labeling to AMPARs occurred at cell-surface using the CAM2(TCO) reagent.

Modification in the main text

In page 7, line 13–: With regards to the molecular weight of the labeled band, the anti-FI signal corresponded to the highest signal among multiple bands that were detected using anti-GluA2/3 antibodies (**Fig. 2b**). The multiple GluA2 bands converged into a single lower band after treatment with peptide-*N*-glycosidase F (PNGase F), which is consistent with previous reports showing that GluA2 is highly glycosylated with *N*-linked sugars³⁸. Importantly, in the PNGase F-treated samples, the shifted anti-GluA2 band merged with the anti-FI signal (**Fig. 2b**). These findings indicate that highly glycosylated GluA2 is selectively labeled using our methods.

In page 11, line 13–: As observed with the AMPARs expressed in HEK293T cells (**Fig. 2a**), smeared bands were detected using anti-GluA2 antibodies; the anti-FI band corresponded to the highest band in the smeared anti-GluA2 signals. After the removal of *N*-linked sugars by PNGase treatment, the smeared anti-GluA2 bands converged into a single lower band, which merged with the anti-FI signal (**Supplementary Fig. 11**). These results suggest that the highly glycosylated fraction of endogenous AMPARs were selectively labeled with fluorescein by the rapid labeling.

Modification in Figures or Supplementary information

- Figure 2b: Deglycosylation assay of GluA2 in HEK293T cells was newly added.
- Supplementary Figure 11: Deglycosylation assay of GluA2 in neurons was newly added.
- Supplementary Figure 17a: Surface-biotin-labeling of GluA2 in neurons was newly added.

Comment 4-2

Can the author calculate the Pearson's correlation coefficient for the merged fluorescence images (e.g., Fig 3c, 4c)?

Our response

As suggested by this reviewer, Pearson's correlation coefficient values were added in the merged fluorescence images.

Modification in Figures or Supplementary information

- Figure 3c: Pearson's correlation coefficient values were added.
- Figure 4c: Pearson's correlation coefficient values were added.

Comment 4-3

Please show anti-GluA2/3 blots for Fig 3e,f,h,i.

Our response

As suggested by this reviewer, we added anti-GluA2/3 blots in Fig 3e, 3f, 3h, 3i (current Supplementary Fig. 18). Similarly, we added anti-GluN2A blots in Fig 6c.

Modification in Figures or Supplementary information

- Figure 3e, f, h: anti-GluA2/3 blots were added as the loading control.
- Figure 6c: anti-GluN2A blot was added as the loading control.
- Supplementary Figure 18: anti-GluA2/3 blot was added as the loading control.

Comment 4-4

Specificity of CNM(TCO) to GluN2A. I can't find the data demonstrating that CNM(TCO) is not reactive to GluN1 or other NMDARs.

Our response

We examined specificity of CNM(TCO) in HEK293T cells transiently expressed with each NMDAR subunit. As shown in Supplementary Figure 21, CNM(TCO) selectively binds to GluN2A among NMDAR family including GluN1, GluN2A, GluN2B and GluN3A.

Modification in the main text

In page 15, line 22–: The 180 kDa band **was not detected** under the control conditions (see lanes 2–5 in **Fig. 6b and Supplementary Fig. 21**), suggesting that this band corresponds to labeled GluN2A.

Modification in Figures or Supplementary information

- Supplementary Figure 21: Subunit selectivity of CNM(TCO) labeling in HEK293T cells was newly added.

Comment 4-5

Include the concentration of CAM2 reagents and tetrazine probes used in all figure captions.

Our response

As suggested by this reviewer, we added the concentration of LDAI reagents (CAM2(TCO) or CNM(TCO)) and tetrazine probes used in all figure captions.

Comment 4-6

Statistical analysis. Please include the number of biological- and/or technical replicates used to calculate average and error values. This information is missing in most supplementary figures.

Our response

As suggested by this reviewer, we added description about the number of samples in each figure legend including supplementary figures. Most experiments were conducted triplicate or more in biologically distinct samples, which is clearly described in the revised manuscript. In addition, both cell numbers and biological replicates were described in Supplementary Figure 8 and 15 legend.

Comment 4-7

Please insert the relevant references to Line 23 of page 11.

Our response

As suggested by this reviewer, we added the corresponding references in this part.

Modification in the main text

In page 13, line 8–: Molecular biology or biochemical methods, such as the genetic incorporation of fluorescent proteins, surface biotinylation assays and metabolic incorporation of radioisotopes, have revealed the diffusion dynamics⁴², recycling process⁵ and half-life^{43,44} of AMPARs, respectively.

Response to Reviewer 3's comments

In this manuscript, Ojima K. et al. developed an improved approach to label endogenous AMPA receptors with small fluorophores in live cells. Building on their earlier success in labeling AMPARs with LDAO, the authors further incorporated an IEDDA reaction into the process. The advantage of IEDDA reaction is its fast kinetics. This allowed faster labeling under more physiological conditions. The authors applied this method to measure the half-life of AMPARs in HEK 293 cells and cultured neurons. They also developed similar labeling method for NMDARs. A method to label endogenous AMPAR with small fluorophores under physiological conditions is appealing to scientists in this research field. The approach presented in this study is an incremental advancement from their earlier work (Wakayama S. et al., 2017). While interesting, there are concerns about unsubstantiated claims and flawed analysis as follows. In addition the authors state that their study revealed some unique features of AMPARs, such as a long lifetime and a rapid recycling in neurons

Our response

We would like to thank the reviewer for kind review and for important comments. According to the suggestions and comments, we have carefully amended our manuscript as shown below. All the revisions we made are highlighted in red in the revised manuscript.

Comment 1

1. The major claim of this manuscript is that the chemical labeling is conducted under physiological conditions. I found this claim is unsubstantiated and overstated.

Our response

As suggested by this reviewer, we modified overstated phrases into more appropriate words or sentences. For example, “chemical labeling is conducted under physiological conditions” was changed into “chemical labeling is conducted under the physiological temperature in culture medium” in this revised manuscript.

Modification in the main text

In page 2, lines 9–10

In page 5, line 14

In page 6, line 14

In page 7, line 27

In page 8, lines 2–3 and line 15

In page 9, lines 13–14

In page 13, line 14

In page 14, line 26

In page 15, line 1

In page 17, line 20 and lines 22–23

In page 19, line 21

Comment 1a

a. For the first labeling step in neurons, a conjugate of PFOX, a potent AMPAR antagonist, was applied for 10 hr. at a concentration over 10-fold higher than Ki. This may induce homeostatic plasticity in neurons and change the trafficking of AMPARs. The authors should check whether the application of CAM2(TCO) affected the physiology of neurons, including amount of surface and synaptic AMPARs and NMDARs, phosphorylation of AMPAR, NMDAR, CREB and ERK. For the case of NMDAR and Pzf, the same should be examined.

Our response

As suggested by this reviewer, we examined influence of **CAM2(TCO)** labeling to surface and synaptic AMPARs and NMDARs, phosphorylation of AMPAR, NMDAR, CREB and ERK. As shown in Supplementary Figure 17a, surface biotinylation assay indicated that amount of surface AMPARs and NMDARs was not affected by **CAM2(TCO)** labeling for 10 hr. Similarly, amounts of synaptic fraction of AMPARs and NMDARs were not affected by the **CAM2(TCO)** labeling (**Supplementary Fig. 17b**). With regards to phosphorylation level of AMPAR, NMDAR, CREB and ERK, constitutive phosphorylation of GluA1 (S831) and GluN1 (S890) was not prominently affected by the **CAM2(TCO)** treatment (**Supplementary Fig. 17d**). However, constitutive phosphorylation of CREB and ERK was not significantly but slightly decreased by **CAM2(TCO)** labeling for 10 h. Considering the high affinity of **CAM2(TCO)** for AMPAR, this influence may be reduced when neurons are treated with low concentration of **CAM2(TCO)**.

With regards to the effect of NMDAR labeling using **CNM(TCO)**, surface and synaptic AMPARs and NMDARs, phosphorylation of AMPAR, NMDAR, CREB and ERK were not prominently affected (**Supplementary Fig. 23**).

These results indicate that most of neuronal function were not affected by our AMPAR or NMDAR labeling. However, homeostatic phosphorylations of synaptic proteins may be partially affected by **CAM2(TCO)** treatment. Thus, we avoid to use the phrase “physiological condition” in this revised manuscript as described above.

Modification in the main text

In page 14, line 1–: Besides, we examined the effects of **CAM2(TCO)** labeling on neuronal function. Neither the surface amount nor the synaptic fraction of AMPARs and NMDARs were affected by **CAM2(TCO)** labeling for 10 h (**Supplementary Figure 17a,b**). Association between GluA2 and the accessory protein, TARPy8 (ref. 45) was also unaffected by **CAM2(TCO)** labeling (**Supplementary Figure 17c**). Although the homeostatic phosphorylation of ERK and CREB was not significantly but slightly decreased, phosphorylation levels of AMPAR (GluA1) and NMDAR (GluN1) were not influenced by the **CAM2(TCO)** labeling (**Supplementary Figure 17d**).

In page 16, line 10–: Importantly, **CNM(TCO)** labeled NMDAR with minimal disturbance to neuronal functions, including the constitutive phosphorylation of ERK and CREB (**Supplementary Fig. 23**).

In page 17, line 5–: The present investigation also revealed that the homeostatic phosphorylation of ERK and CREB was slightly decreased by **CAM2(TCO)** but not **CNM(TCO)** treatment. Considering the

high affinity of CAM2(TCO) for AMPAR, this influence may be reduced when neurons are treated with low concentration of CAM2(TCO).

Modification in Figures or Supplementary information

- Supplementary Figure 17: Influence of CAM2(TCO) labeling to neuronal functions was newly added.
- Supplementary Figure 23: Influence of CNM(TCO) labeling to neuronal functions was newly added.

Comment 1b

b. Although the first step is conducted under physiological condition (in culture medium at 37 °C), the 2nd step is performed under non-physiological condition (in PBS at room temperature). As we know, solution exchange from culture medium to salt solution acts as a stimulus to neurons. This is especially problematic since all their measurements of half-time, intracellular, surface or recycled percentages of AMPARs assume that surface and intracellular AMPARs are in equilibrium and not affected by the labeling process. Thus, this non-physiological step calls into question of those measurements.

Our response

To respond to this reviewer's comment, we conducted both CAM2(TCO) labeling and Tz(FI) labeling in culture medium at 37 °C for determining the surface and intracellular ratio of AMPARs. As shown in Supplementary Figure 19c, the ratio was comparable to those obtained under the original condition in which Tz(FI) labeling was conducted in PBS for 5 min (Fig. 5c). We believe that trafficking of AMPARs would not be prominently affected in our analyses. However, our current protocols include non-physiological step as commented by this reviewer. Thus, we avoid to use the phrase "physiological condition" in the revised manuscript as described above.

Modification in Figures or Supplementary information

- Supplementary Figure 19c: We newly added the data for tetrazine ligation in cell culture medium.

Comment 2

2. The definition of half-time of surface AMPARs is ambiguous. And the determination of this parameter is flawed.

Comment 2a

a. HEK293T cells double about every 24-30 hr. The authors did not take this in consideration when analyzing half-time of surface AMPARs in HEK293T cells with confocal imaging.

Our response

As pointed out by this reviewer, HEK293T cells double about every 24-30 hr. This means that labeled AMPARs are reduced by almost half in the cell division step. This means that it is difficult to analyze the labeled AMPAR quantitatively in the long period by confocal imaging. Thus, in this revised manuscript, we analyzed the half-time of surface AMPARs within 8 h after the Tz(Ax488) labeling by confocal imaging.

Modification in the main text

In page 9, line 13–: After incubating the cells with CAM2(TCO) under physiological temperature in

culture medium, Tz(Ax488) was added to the culture medium to selectively visualize cell-surface AMPARs and cells were incubated for each period (0–8 h) (Fig. 3a).

Modification in Figures or Supplementary information

- Figure 3b: The time-course of the confocal imaging was changed from 0–24 h to 0–8 h.

Comment 2b

b. For WB analysis in both neurons and HEK293T cells, cells were incubated with CAM2(TCO) for 4 or 10 hr., during which time some fraction of surface and intracellular AMPARs were labeled with TCO. During the incubation period of 0-72 hr., TCO-AMPARs are under constant recycling. The measured Tz(Fl)-labeled TCO-AMPARs contain two populations: those that remained on the cell surface since time 0; and those that were recycled back to surface. The measurement of half-time of surface AMPARs is complicated by this 2nd population.

Our response

In the analyses, we would like to know the time-dependency of cell-surface AMPARs which include both the resident and recycled fraction. However, as pointed out by this reviewer, the definition, half-life of surface AMPARs ($t_{1/2}^{\text{surface}}$) was ambiguous in our original manuscript. To improve this issue, we clearly described our definition of the $t_{1/2}^{\text{surface}}$ which includes both remaining and recycled AMPAR on the surface in the main text.

Modification in the main text

In page 9, line 19–: The half-time of cell-surface AMPARs ($t_{1/2}^{\text{surface}}$), which includes both the remaining and recycled fractions, was calculated to be 5.7 ± 0.7 h from the fluorescent intensity on the cell surface (Fig. 3b).

In page 14, line 15–: We therefore applied biochemical methods for analyzing the $t_{1/2}^{\text{surface}}$ of AMPARs, including remaining and recycled components (Fig. 5a).

Comment 3

3. They mentioned in the discussion that half-lives of synaptic proteins are controversial. Previous work has come to similar conclusions for AMPAR half-lives and rapid recycling so it is not clear how it is controversial. It is also not clear how the new approach resolved the conflicts. Due to the flaws with current analysis, it may add more confusion.

Our response

As described in our original manuscript, previous studies using radiolabeling methods indicated the half-lives of synaptic proteins are 1–2 days. In contrast, other groups reported that these proteins are decomposed within several hours in cultured neurons. So, we added the sentence “the half-lives of synaptic proteins are currently controversial” in the “Discussion” section in the original manuscript. However, the main point of this paragraph (line 23 in page 18 in the revised manuscript) is that molecular

mechanisms of AMPAR trafficking between neurons and HEK293T cells. Considering both the reviewer's comment and the main point of this paragraph, the description of the controversial issue with respect to the previous report would not be required in this paragraph. Thus, in the revised manuscript, we removed the sentence "the half-lives of synaptic proteins are currently controversial", and focused on the molecular mechanisms of AMPAR trafficking between HEK293T cells and neurons. Thank you for your helpful comment.

Modification in the main text

In page 18, lines 25–: **Previous studies using radiolabeling methods indicated that the half-lives of synaptic proteins are 1–2 days^{43,44}. Consistent with previous results, our present investigation demonstrated that the half-lives of AMPARs and NMDARs are 33.2 h and 22.6 h, respectively, in cultured neurons.**

Comment 4

4. The labeling of NMDARs shows much higher background than AMPARs (Figure 6bd). There is labeling even in the absence of TCO (lane 5 in Figure 6b). This indicates this method is case-dependent and limits the general application to other receptors.

Our response

We noticed that original **CNM(TCO)** was partially hydrolyzed in the DMSO stock solution. Then, we re-purified the **CNM(TCO)** by HPLC, and we examined same experiment to confirming the selectivity. As shown in revised Figure 6b and 6d, background signals were weakened, and strong signal corresponding to labeled NMDAR was observed clearly. **CNM(TCO)** is hydrophilic and susceptible to hydrolysis even in DMSO solution. Thus, in the revised manuscript, we described this point in method section of two step labeling of AMPARs and NMDARs. However, the background level is still higher, compared with those obtained in the AMPAR labeling using **CAM2(TCO)**. The main reason would be low affinity of **CNM(TCO)** to NMDAR compared with the affinity of **CAM2(TCO)** to AMPARs. To confirm this consideration, we examined concentration dependency of the chemical labeling of AMPARs or NMDARs using **CAM2(TCO)** or **CNM(TCO)**, respectively in this revision (**Supplementary Fig. 13b** and **22b**). We calculated the EC_{50} value of the **CAM2(TCO)** labeling as $0.90 \pm 0.10 \mu\text{M}$ to AMPARs in neuron (**Supplementary Fig. 13b**). In contrast, in the case of **CNM(TCO)** labeling to NMDARs, saturation of the labeled band is not observed in this concentration range ($[\text{CNM(TCO)}] = 0 - 10 \mu\text{M}$) (**Supplementary Fig. 22b**). These results may indicate that one of the next direction would be utilization of high-affinity ligands to NMDAR to improve the selectivity. We will perform our research in this direction and would like to report them in the future. Anyway, we believe that the present two-step labeling have a potential to other neurotransmitter receptors.

Modification in the main text

In page 12, line 6–: **In addition, the concentration dependency of CAM2(TCO) revealed the EC_{50} value ($0.90 \pm 0.10 \mu\text{M}$) of two-step labeling at 37 °C in neurons (Supplementary Fig. 13b).**

In page 16, line 6–: **However, the concentration dependency of CNM(TCO) for NMDAR labeling was**

different from that of **CAM2(TCO)** for AMPARs. As shown in Supplementary Figure 22b, the labeled bands were not saturated in the 0–10 μM range, indicating that the affinity of **CNM(TCO)** was lower than that of **CAM2(TCO)**.

In page 21, line 23–: **CAM2(TCO)**, **CNM(TCO)** and Tz probes were stored in DMSO solution. These were kept in deep freezer ($-80\text{ }^{\circ}\text{C}$) to prevent decomposition, because these compounds were not stable in DMSO at ambient temperature.

Modification in Figures or Supplementary information

- Figure 6b and 6d: The data was revised using freshly prepared **CNM(TCO)**.
- Supplementary Fig. 13b: Concentration dependency of **CAM2(TCO)** for AMPAR labeling in neurons was newly added.
- Supplementary Fig. 22b: Concentration dependency of **CNM(TCO)** for NMDAR labeling in neurons was newly added.

Comment 5

5-1. What's the cell permeability of CAM2(TCO)?

Our response

Considering the anionic and hydrophilic feature of **CAM2(TCO)**, it is expected that **CAM2(TCO)** has not permeability into cells. In addition, we previously showed that LDAI chemistry is not suitable for chemical labeling to intracellular proteins mainly due to low stability of the labeling reagent under the intracellular condition (Takaoka *et al.*, *Chem. Sci.* **6**, 3217–3224 (2015)). Moreover, in this revision, we evaluated **CAM2(TCO)** labeling using cell lysate containing intracellular components. As shown in the attached Figure B (see below), two-step labeling using **CAM2(TCO)** failed to selectively labeling to AMPARs. In addition, other experiments including surface biotinylation assay revealed that **CAM2(TCO)** labeled AMPARs on cell-surface (for details, see our response to your *Comment 14*). Thus, we believe that **CAM2(TCO)** does not permeate into cells.

Figure B. Two-step labeling of cell-surface AMPARs on live HEK293T cells or in the cell lysate. The live cells or the cell lysate were incubated with 2 μM **CAM2(TCO)** for 4h followed by the addition of 1 μM **Tz(FI)** for 5min at 37 $^{\circ}\text{C}$ in the presence or absence of 50 μM NBQX. Western blots using anti-FI was shown.

5-2. What percentage of surface and intracellular receptors in HEK293 cells and neurons were labeled under the conditions used?

Our response

As suggested by this reviewer, we determined how efficiently labeled the native AMPARs in neurons by quantifying the unlabeled AMPARs after the two-step labeling using western blotting.

Prior to the quantification by western blotting, we needed to focus on the existence of several anti-GluA2 bands, which was pointed out by this reviewer (*Comment 14*) and reviewer #2 (*Comment 4*). In this revision, we revealed that these bands were caused by different glycosylation patterns of GluA2 (see **Supplementary Fig. 11**). In addition, we confirmed that the highest band corresponded to cell-surface GluA2 using surface-biotin-labeling (see **Supplementary Fig. 17a**).

Based on these results, we quantified the remaining unlabeled GluA2 fraction by focusing the highest GluA2 band in neuron. In this experiment, after two-step labeling, FI-labeled AMPARs were immunoprecipitated from the cell lysate using anti-fluorescein antibody. As shown in Supplementary Figure 12a, $44 \pm 4\%$ of GluA2-containing AMPARs were recognized by our present labeling method. Similarly, we calculated that $37 \pm 7\%$ of GluA1- and $43 \pm 5\%$ of GluA3-containing AMPARs were recognized. With regards to GluA4, another AMPAR subunit, western blotting using anti-GluA4 antibody failed to detect GluA4 in our primary cortical culture, which is consistent with previous reports (e.g. ref. 41) showing low level expression of GluA4 in cortical neuronal culture.

However, considering the hetero-tetrameric formation of AMPAR subunits in neurons, we need to confirm whether each subunit is covalently labeled with the probe or not. To address this issue, after two-step labeling using **CAM2(TCO)** and **Tz(FI)**, we examined immunoprecipitation assay by anti-fluorescein antibody using the cell lysate in the denatured condition under which tetrameric formation of AMPARs is collapsed. As shown in Supplementary Figure 12b, the immunoprecipitation assay revealed that **CAM2(TCO)** was covalently labeled to GluA2 or GluA3 but not to GluA1. This selectivity is consistent with our previous results showing covalent labeling to GluA2, GluA3 or GluA4 among AMPAR subunits (GluA1–4) in HEK293T cells (ref. 26).

The labeling efficiency to GluA2 was similarly examined in HEK293T cells by quantifying the unlabeled AMPARs after the two-step labeling using western blotting. As shown in Supplementary Figure 7, we revealed that $35 \pm 3\%$ of tetrameric AMPARs were recognized by our present labeling method.

These results indicate that all of AMPARs are not chemically labeled in our method. However, we think that it is not problematic in the analyses of AMPARs in neurons because sufficient fluorescent intensity or band intensity is detected in confocal imaging or western blotting, respectively.

Modification in the main text

In page 9, line 3–: **With regards to labeling efficacy, quantification of the remaining unlabeled GluA2 fraction showed $35 \pm 3\%$ of surface AMPARs were visualized in the two-step labeling (Supplementary Fig. 7).**

In page 11, line 13–: As observed with the AMPARs expressed in HEK293T cells (**Fig. 2a**), smeared bands were detected using anti-GluA2 antibodies; the anti-FI band corresponded to the highest band in the smeared anti-GluA2 signals. After the removal of *N*-linked sugars by PNGase treatment, the smeared anti-GluA2 bands converged into a single lower band, which merged with the anti-FI signal (**Supplementary Fig. 11**). These results suggest that the highly glycosylated fraction of endogenous AMPARs were selectively labeled with fluorescein by the rapid labeling.

Of the AMPAR subunits (GluA1–4), GluA1, GluA2 and GluA3 are highly expressed in cultured cortical neurons⁴¹. We next examined the efficacy of our methods for visualizing tetrameric AMPARs by quantifying the remaining unlabeled GluA2 fraction. As shown in Supplementary Figure 12a, $44 \pm 4\%$ of GluA2-containing AMPARs were recognized by the two-step labeling method. Similarly, we calculated that $37 \pm 7\%$ of GluA1- and $43 \pm 5\%$ of GluA3-containing AMPARs were recognized. However, considering the heterotetrameric formation of AMPAR subunits, we also needed to examine whether each subunit was covalently labeled with the probe or not. In this context, the immunoprecipitation assay in the denatured condition revealed that GluA2 and GluA3, but not GluA1, were covalently labeled with **CAM2(TCO)** (**Supplementary Fig. 12b**). This selectivity is consistent with our previous results²⁶ in HEK293T cells.

Modification in Figures or Supplementary information

- Supplementary Figure 7: Labeling efficacy and subunit selectivity of the labeling in HEK293T cells were newly added.
- Supplementary Figure 11: Deglycosylation assay of GluA2 in neuron was newly added.
- Supplementary Figure 12: Labeling efficacy and subunit selectivity of the labeling in neurons were newly added.
- Supplementary Figure 17a: Surface-biotin-labeling of GluA2 in neuron was newly added.

5-3 Does the label affect the channel property, post-translational modifications, or interaction with other associated proteins?

Our response

We previously confirmed that **CAM2(FI)** labeling did not affect the concentration dependency of glutamate or ion-channel kinetics of AMPARs by electrophysiological assay (ref. 26: Wakayama *et al. Nat. Commun.* 2017). With regards to **CAM2(TCO)** utilized in this paper, the linker length between ligand group and reactive acyl imidazole group is completely same with that of **CAM2(FI)**. Thus, it is expected that two-step labeling using **CAM2(TCO)** has little effect on AMPAR function. In addition, in this revision, we evaluated effect of two-step labeling using **CAM2(TCO)/Tz(FI)** pair or **CAM2(TCO)/Tz(Ax488)** pair on the AMPAR response. As shown in Supplementary Figure 8, Ca^{2+} imaging indicated that two-step labeling did not prominently affect the concentration dependency of glutamate in HEK293T cells transiently expressed with GluA2.

With regards to post-translational modification, we revealed that GluA2 is highly glycosylated,

and CAM2(TCO) selectively labeled highly glycosylated GluA2 on the cell-surface in this revision (**Supplementary Fig. 11**). Importantly, the band pattern of anti-GluA2 was not influenced by CAM2(TCO) treatment, which indicates that glycosylation of GluA2 is not affected by CAM2 labeling. This result is consistent with the effect of CAM2 labeling on the surface labeling of GluA2 as described above (**Supplementary Fig. 17a** and see our response to your comment #1a).

AMPA is well known to associate with TARP γ 8 in hippocampal neurons. Then, we evaluated effect of CAM2(TCO) labeling to these association by immunoprecipitation. As shown in Supplementary Figure 17c, TARP γ 8 was co-immunoprecipitated using anti-GluA2 antibody, which was unaffected by CAM2(TCO) treatment (**Supplementary Fig. 17c**).

These results indicate that we can analyze AMPAR trafficking with negligible disturbance of receptor function using the two-step labeling.

Modification in the main text

In page 9, line 5–: Moreover, AMPAR function was not visibly affected by the two-step labeling (**Supplementary Fig. 8**), which is consistent with our previous analyses that showed minimal disturbance of AMPAR ion channel properties by CAM2 labeling²⁶.

In page 11, line 13–: As observed with the AMPARs expressed in HEK293T cells (**Fig. 2a**), smeared bands were detected using anti-GluA2 antibodies; the anti-FI band corresponded to the highest band in the smeared anti-GluA2 signals. After the removal of *N*-linked sugars by PNGase treatment, the smeared anti-GluA2 bands converged into a single lower band, which merged with the anti-FI signal (**Supplementary Fig. 11**). These results suggest that the highly glycosylated fraction of endogenous AMPARs were selectively labeled with fluorescein by the rapid labeling.

In page 14, line 4–: Association between GluA2 and the accessory protein, TARP γ 8 (ref. 45) was also unaffected by CAM2(TCO) labeling (**Supplementary Figure 17c**).

Modification in Figures or Supplementary information

- Supplementary Figure 8: Effects of the two-step labeling to AMPAR function were newly added.
- Supplementary Figure 11: Deglycosylation assay of GluA2 in neuron was newly added.
- Supplementary Figure 17c: Influence of CAM2(TCO) labeling to association between GluA2 and TARP γ 8 was newly added.

Comment 6

6. When comparing the % recycled in neurons and HEK293T cells, they used two different labeling schemes (Figure 5 and Sp. Figure 11), e.g. the order of Tz(Ax647) and Tz(Fl). Due to the possibility that the reaction kinetics and background are different for the two dyes, the same scheme should be used for direct comparison.

Our response

As suggested by this reviewer, we applied the same labeling scheme between neurons and HEK293T cells. In other words, labeled AMPARs were quantified using anti-Alexa647 antibody in HEK293T cells like those obtained in neurons. This data was added as Supplementary Figure 20.

Modification in Figures or Supplementary information

- Supplementary Figure 20: The quantification of surface and intracellular labeled AMPAR using anti-Alexa647 antibody in HEK293T cells was newly added.

Comment 7

7. Do all Tz reagents (Ax488, Fl, Ax647) have similar reaction kinetics? The authors should provide these data.

Our response

As requested by this reviewer, we examined reaction kinetics of **Tz(Fl)** and **Tz(Ax647)** in addition to **Tz(Ax488)**. As shown in Supplementary Figure 6c, reaction kinetics of these probes (**Tz(Ax488)** or **Tz(Ax647)**) showed similar reaction kinetics with that of **Tz(Fl)**.

Modification in Figures or Supplementary information

- Supplementary Figure 6c: Reaction kinetics of **Tz(Ax488)** or **Tz(Ax647)** was newly added.

Comment 8

8. What is the meaning of n numbers in the manuscript?

Our response

As suggested by this reviewer, we added description about the number of samples in each figure legend including supplementary figures. Most experiments were conducted triplicate or more in biologically distinct samples, which is clearly described in the revised manuscript. In addition, both cell numbers and biological replicates were described in Supplementary Figure 8 and 15 legend.

Comment 9

9. Figure 2a: The 70-kDa band is still present although weak. The authors should quantify the intensity.

Our response

As suggested by this reviewer, we quantified the 70 kDa band, a major nonspecific band mainly observed in **CAM2(Fl)** labeling in serum-containing medium in Supplementary Figure 3.

Modification in the main text

In page 7, line 20–: **Furthermore**, in the case of direct fluorescein labeling using the original **CAM2(Fl)** under the same conditions (see **Supplementary Fig. 1b** for its structure), there was a strong band around 70 kDa as well as the 110 kDa band (lanes 6–7 in **Fig. 2a** and **Supplementary Fig. 3**).

Modification in Figures or Supplementary information

- Supplementary Figure 3: The quantification of the band intensity for both 70 and 110 kDa bands was newly added.

Comment 10

10. Figure 3b and 5b: the curve does not plateau at 0. Why is that?

Our response

In the original manuscript, offset value of the fit function was not set equal to zero. The reason was that our original manuscript lacked the data where most of labeled AMPARs were decomposed or internalized. In this revision, we re-evaluated the internalization or degradation analyses by incubating the cells for longer periods (0 – 36 h in HEK cells or 0 – 120 h in neurons). In addition, the non-specific band was observed in original Figure 5b (NMDAR labeling in HEK293T cells), which is another reason why the curve does not plateau at 0 in this figure. As mentioned in your *comment 4*, non-specific band is prominently decreased in the present Figure 5b. As a result, offset value could be set equal to zero in new data. Importantly, although the newly obtained $t_{1/2}$ values were slightly different from original values, the tendency and our claim were not changed at all.

Modification in the main text

In page 23, line 12–: The membrane intensity was fitted with KaleidaGraph using following equation : $F = a + b \cdot e^{-ct}$, and the offset value (a) was set equal to zero. The $t_{1/2}$ was defined as $t_{1/2} = \ln(2)/c$.

In page 24, line 9–: The half-life was calculated by curve fitting using KaleidaGraph and following equation: $I = a + b \cdot e^{-ct}$, and the offset value (a) was set equal to zero. The $t_{1/2}$ was defined as $t_{1/2} = \ln(2)/c$.

Modification in Figures or Supplementary information.

- Figure 3b: The offset value (a) was set equal to zero using the original data.
- Figure 3e and 3f: The data was revised by changing the incubation time from 0–24 h to 0–36 h.
- Figure 5b: The data was revised by changing the incubation time from 0–72 h to 0–120 h.
- Figure 6c: The data was revised by changing the incubation time from 0–24 h to 0–36 h.
- Figure 6e: The data was revised by changing the incubation time from 0–72 h to 0–120 h.

Comment 11

11. Figure 3h: the band for “Surface” does not look like 94% of the “Whole”. And there are multiple bands for “Intracellular” lane. How was it quantified?

Our response

In the original Figure 3h, we showed not representative one. In this revision, we selected a more representative data in the revised Figure 3h. With regards to quantification, we subtracted the background signal from the target band. The background signal was selected from the same lane not including any obvious signals. In this revision, this is clearly described in method section.

Modification in the main text

In page 25, line 4–: The target bands were manually selected, and the intensity were calculated with ImageJ software, background intensity was manually subtracted by selecting a region with no bands from the same lane. In more detail, the band intensity was determined as described below.

$$(\text{target intensity}) - (\text{target area}) / (\text{background area}) \times (\text{background intensity})$$

Comment 12

12. It is not clear how the enrichment of surface AMPARs in spines vs. dendrites were quantified (Figure 4d).

Our response

We added the description how to quantify the enrichment of surface AMPARs in spines vs. dendrites in the method section.

Modification in the main text

In page 29, line 26–: FLIM images were processed in LAS X 3.5.5 software (Leica microsystems) to fit the lifetime decay curves using an n-exponential reconvolution model with the number of components that χ^2 value is closest to 1. In our data, a three-component fit was utilized. The component of the lifetime corresponding to SeTau647 ($\tau = 2.4 \pm 0.1$ ns) was used for calculating the intensity of FLIM images. The synaptic or dendric region was selected ROI on PSD95 or MAP2 signals and the background intensities were subtracted by selecting a region of no cells.

Comment 13

13. How neuronal viability was quantified (Sp. Figure 8 and 9)?

Our response

We added the description how to quantify the neuronal viability in Supplementary Method section.

Modification in Figures or Supplementary information

SUPPLEMENTARY INFORMATION

In page 29, line 4–: Calcein positive cells were regarded as live cells, and total cell number were counted by Hoechst 33342 fluorescence. The cell viability rate (live cell / total cell) was calculated according to the following formula: (Number of Calcein positive cells) / (Number of Hoechst33342 positive cells). The images were obtained four independent experiments each (n = 22 – 101 cells). The ROI of these cells were manually selected using ImageJ software.

Comment 14

14. Sup. Figure 6b and 10b: Why are there two band using anti-GluA2, but single band with anti-F1?

Our response

Previous studies have revealed that GluA2 undergo the posttranslational modification of N-linked glycosylation (e.g. ref. 38). Thus, a plausible explanation of the difference is that these bands correspond to the different glycosylation pattern of GluA2. To examine this possibility, we treated cell lysate with peptide-N-glycosidase F (PNGase F) for removal of all N-linked sugars. As shown in Supplementary Figure 11, after treatment with PNGase F, western blotting using anti-GluA2 showed single band which shifted to the lower molecular weight direction. This indicates that these bands in anti-GluA2 correspond to different glycoform pattern of GluA2.

With regards to detection of one band by anti-fluorescein antibody, molecular weight of the label

band corresponds to the highest band in anti-GluA2 (see **Supplementary Fig. 11**). After treatment of PNGase F to these samples, molecular weight of the anti-FI band shifted toward lower molecular weight, and the molecular weight was comparable to that of the deglycosylated GluA2 band (**Supplementary Fig. 11**). These indicate that the highest band in anti-GluA2 is selectively labeled by our methods. In addition, surface biotinylation assay revealed that the highest band corresponds to cell-surface GluA2 (**Supplementary Fig. 17**). These results support selective labeling to AMPARs occurred at cell-surface using the CAM2(TCO) reagent.

Modification in the main text

In page 11, line 13–: *As observed with the AMPARs expressed in HEK293T cells (Fig. 2a), smeared bands were detected using anti-GluA2 antibodies; the anti-FI band corresponded to the highest band in the smeared anti-GluA2 signals. After the removal of N-linked sugars by PNGase treatment, the smeared anti-GluA2 bands converged into a single lower band, which merged with the anti-FI signal (Supplementary Fig. 11). These results suggest that the highly glycosylated fraction of endogenous AMPARs were selectively labeled with fluorescein by the rapid labeling.*

Modification in Figures or Supplementary information

- Supplementary Figure 11: Deglycosylation assay of GluA2 in neurons was newly added.
- Supplementary Figure 17a: Surface-biotin-labeling of GluA2 in neurons was newly added.

Comment 15

15. _____ p. 16, line 17-20: Need references.

Our response

As suggested by this reviewer, we added the appropriate references.

Modification in the main text

In page 18, line 25: *Previous studies using radiolabeling methods indicated that the half-lives of synaptic proteins are 1–2 days^{43,44}.*

Comment 16

16. _____ p. 17 line 8-9: Need references.

Our response

As suggested by this reviewer, we added the appropriate references.

Modification in the main text

In page 19, line 15: *However, in some cases, complementary genetic experiments using knock-in or knock-out mice of the target gene have not supported the data¹⁹.*

REVIEWERS' COMMENTS

Reviewer #1 (Remarks to the Author):

I am very impressed by the revised manuscript of Ojima et al. on affinity-based fluorescence labeling of glutamate receptors in live neuronal cells. They have addressed the main points of concern and have made valuable additions to the manuscript. Therefore, I highly recommend publication of this work.

Reviewer #2 (Remarks to the Author):

The authors have addressed all my concerns.

Reviewer #3 (Remarks to the Author):

In this revised version, the authors performed additional experiments to address the reviewers' comments, and revised the text to make appropriate claims. The presented labeling approach will be a valuable tool to the scientific community especially if the chemicals could be made widely available to other researchers. I would recommend its publication.